# Enzyme-mediated aminoglycoside resistance without target mimicry
**Mark Hemmings** [1,2,3], **Michał Zieliński**[1,2], **Tolou Golkar**[1,2,3], **Jonathan Blanchet**[1,2,3], **Angelos Pistofidis**[1,2,3], **Kim Munro**[2], **T. Martin Schmeing** [1,2], **D. Scott Bohle**[4] **& Albert M. Berghuis** [1,2,3,5] ✉

The primary mode of resistance to aminoglycoside antibiotics is through chemical modification catalyzed by aminoglycoside-modifying enzymes. Numerous structural studies of these enzymes have invariably shown that they bind aminoglycosides in the same lowest-energy conformation as the intended target for these antibiotics, the A site of the bacterial ribosome. Presumably, the binding mode mimicry enables these enzymes to compete successfully with the target, thus conferring effective resistance. Here we present the first structural and functional studies of two aminoglycoside-modifying enzymes that do not use target mimicry, AAC(3)-Ia and AAC(3)-XIa. X-ray diffraction studies reveal that these enzymes bind aminoglycoside antibiotics in a conformation where the central 2-deoxystreptamine ring is in boat conformation. The effect of this non-canonical binding mode on the enzymes' ability to modify antibiotics is assessed in silico and in vitro, and its impact for conferring resistance is assessed in vivo. Overall, the results show that target mimicry, while advantageous, is not an essential strategy for aminoglycoside-modifying enzymes to be effective in conferring resistance.

In 2019, antibiotic resistance killed 1.27 million people globally[1]. Widespread resistance has led to a decrease in the clinical use of many antibiotics including aminoglycosides. Aminoglycosides are a broad-spectrum class of antibiotics that are used to treat serious Gram-negative and Gram-positive bacterial infections[2]. There are three major subclasses of aminoglycosides based on their chemical structures[3,4]. The largest of these subclasses, which includes the clinically relevant antibiotics sisomicin, gentamicin, tobramycin, and neomycin, are characterized by a 2-deoxystreptamine (2-DOS) core that can be substituted with amino sugar rings at the 4 and 5 positions, or at the 4 and 6 positions of the central ring (Supplementary Fig. 1)[3]. These antibiotics bind to the bacterial ribosomal A site, resulting in an error-prone protein synthesis that rapidly kills the bacteria[4].

Unfortunately, bacteria have a variety of strategies to circumvent or inactivate aminoglycoside antibiotics[4]. Primarily, bacteria employ aminoglycoside-modifying enzymes to covalently alter these antibiotics, hindering their ability to bind the bacterial ribosome, ultimately protecting themselves from an aminoglycoside's bactericidal effects[5]. The three classes of aminoglycoside-modifying enzymes are aminoglycoside phosphotransferases (APHs), acetyltransferases (AACs), and nucleotidyltransferases (ANTs), which detoxify the antibiotic through the addition of a phosphate, acetyl group, or adenosine monophosphate, respectively[5].

It has been observed that aminoglycoside-modifying enzymes bind aminoglycosides in a low-energy conformation that is essentially identical to the binding mode seen in complexes with their ribosomal target[6–9]. This phenomenon was first noted by the current study's corresponding author in Fong and Berghuis[6], and was termed "target mimicry"[6]. This observed low-energy conformation features the central 2-DOS ring and attached sugar moieties in chair conformations with the vast majority of substituents on the various rings in equatorial positions[10]. Fong and Berghuis speculated that resistance enzymes may have evolved this strategy to better compete with the ribosome[6]. Through evolution, aminoglycoside biosynthetic pathways may have been optimized to produce antibiotics that can bind to the bacterial ribosomal A site in their lowest energy conformation. They postulated that this would be advantageous, as it would enhance aminoglycoside binding affinity by reducing loss of entropy[6]. Similarly, aminoglycoside-modifying enzymes may have evolved to bind this same low-energy aminoglycoside conformation, to improve binding affinity and better compete with the ribosomal target[6]. For the past two decades, target mimicry has not only been invariably observed for aminoglycoside-modifying enzymes, but also for other antibiotic resistance enzymes such as those that target streptogramins and macrolides[7,9].

In this work, we identify the first aminoglycoside-modifying enzymes that do not utilize target mimicry. Crystal structures of AAC(3)-Ia and AAC(3)-XIa reveal aminoglycosides bound with a boat conformation central ring. AAC(3)-Ia was the first plasmid-encoded acetyltransferase

[1]Department of Biochemistry, McGill University, Montréal, QC, Canada. [2]Centre de Recherche en Biologie Structurale, McGill University, Montréal, QC, Canada. [3]Antimicrobial Resistance Centre, McGill University, Montréal, QC, Canada. [4]Department of Chemistry, McGill University, Montréal, QC, Canada. [5]Department of Microbiology and Immunology, McGill University, Montréal, QC, Canada. ✉e-mail: albert.berghuis@mcgill.ca

identified from clinical isolates of *Pseudomonas aeruginosa*, in 1972[11–13]. It provides resistance to gentamicin, sisomicin, and fortimicin[14]. Crystal structures of AAC(3)-Ia in complex with coenzyme A have been published[15,16], but no structures in complex with an aminoglycoside had been previously reported. AAC(3)-XIa was identified in 2015 from *Corynebacterium striatum*[17]. It provides resistance to kanamycin B, tobramycin, dibekacin, gentamicin, sisomicin, and fortimicin[17]. In this study, we investigated target mimicry and whether this strategy is essential to aminoglycoside resistance enzyme success. We compared AAC(3)-Ia and AAC(3)-XIa against the confirmed target-mimicking resistance enzyme AAC(3)-IIIa[18]. Enzyme AAC(3)-IIIa was selected for these experiments because it catalyzes the same acetylation reaction on the aminoglycoside N-3 group as AAC(3)-Ia and AAC(3)-XIa. Specifically, we assessed boat conformation stability through density functional theory calculations, catalytic efficiency through enzymatic assays, thermodynamically characterized aminoglycoside binding through isothermal titration calorimetry, and measured the impact of non-canonical binding on bacterial antibiotic susceptibility.

## Results

### Overall fold and substrate binding to AAC(3)-Ia and AAC(3)-XIa

We have determined two crystal structures of AAC(3)-Ia from *Acinetobacter baumannii* and five of AAC(3)-XIa from *C. striatum*, in complex with various aminoglycosides and either CoASH or AcCoA, as well as a structure where one of two active sites contains AcCoA (Table 1). Both AAC(3)-Ia

structures only contain a single protomer per asymmetric unit. This protomer is in complex with the protomer of a neighboring asymmetric unit to form the physiological dimer. In contrast, all five structures of AAC(3)-XIa contain both protomers of the physiological dimer in the asymmetric unit. The structure we refer to as partial-apo (Table 1) of AAC(3)-XIa contained one protomer devoid of ligands, and the second protomer bound to AcCoA, despite its absence in the crystallization solution, a situation also reported with other AACs[19]. In contrast, all other structures, of either enzyme bound to CoASH or its analog, Ac$_{NH}$CoA, had these ligands present at both active sites in the dimer, regardless of whether this is required by crystallographic symmetry (sisomicin with AAC(3)-Ia; Fig. 1A), or not (AAC(3)-XIa structures; Fig. 1B). The presented structures were determined to 1.4–2.3 Å resolution and associated crystallographic data are provided in Table 1. Similar to other AACs, AAC(3)-Ia and AAC(3)-XIa display the GNAT superfamily fold. They have a pairwise RMSD of 2.7 Å. These enzymes crystallize as C-terminal swapped dimers and display similar structural characteristics to other swapped-dimer AACs, including AAC(6')-Iy (PDB ID: 1S3Z) and AAC(3)-Ib (PDB ID: 4YFJ), with pairwise RMSD between monomeric units of 3.7 and 1.0 Å, respectively for AAC(3)-Ia and 3.4 and 2.7 Å, respectively for AAC(3)-XIa[19].

The partial-apo structure of AAC(3)-XIa reveals no conformational shifts in the enzyme upon CoA binding. The AcCoA binding pockets of both AAC(3)-Ia and AAC(3)-XIa have a similar tunnel-like conformation that extends to the aminoglycoside binding pocket, and our structures show

## Table 1 | Data collection and refinement statistics

| | AAC(3)-Ia + CoASH | AAC(3)-Ia + CoASH + Sisomicin | AAC(3)-Xia partial apo | AAC(3)XIa + acCoA | AAC(3)XIa + CoASH + ac-Sisomicin | AAC(3)XIa + CoASH + Tobramycin | AAC(3)-XIa + ac$_{NH}$CoA + Gentamicin C1 |
|---|---|---|---|---|---|---|---|
| **Data collection** | | | | | | | |
| Space group | P 43 21 2 | P 43 21 2 | P 1 21 1 | P 1 21 1 | P 1 21 1 | P 1 21 1 | P 1 21 1 |
| Cell dimensions | | | | | | | |
| *a, b, c* (Å) | 57.5, 57.5, 123.4 | 57.2, 57.2, 125.2 | 35.2, 99.9, 40.0 | 35.4, 99.5, 39.3 | 35.5, 100.6, 39.3 | 35.2, 101.6, 39.8 | 35.4, 100.6, 39.2 |
| α, β, γ (°) | 90 90 90 | 90 90 90 | β = 106.9 | β = 106.4 | β = 106.6 | β = 106.1 | β = 106.6 |
| Resolution (Å) | 28.9–2.3 (2.67–2.34) | 30.00–2.1 (2.18–2.1) | 29.9–2.1 (2.15–2.08) | 28.5–1.4 (1.43–1.38) | 50.3–1.5 (1.51–1.46) | 32.1–2.0 (2.03–1.96) | 37.5–1.4 (1.42–1.37) |
| $R_{merge}$ | 0.2969 (1.055) | 0.09014 (0.3707) | 0.053 (0.152) | 0.031 (0.266) | 0.046 (0.213) | 0.036 (0.126) | 0.035 (0.175) |
| $I / \sigma I$ | 7.2 (1.98) | 14.1 (1.2) | 15.7 (6.9) | 15.9 (3.2) | 23.1 (6.5) | 18.6 (8.1) | 18.8 (1.1) |
| Completeness (%) | 92.84 (81.7) | 85.07 (100.0) | 99.4 (99.4) | 99.9 (99.8) | 97.7 (94.2) | 99.2 (98.4) | 97.4 (93.0) |
| Redundancy | 8.4 (8.3) | 21.1 (21.4) | 9.5 (9.3) | 10.9 (7.2) | 5.3 (5.3) | 18.5 (7.7) | 3.2 (3.0) |
| **Refinement** | | | | | | | |
| Resolution (Å) | 28.9–2.3 (2.67–2.34) | 30.0–2.1 (2.18–2.1) | 29.9–2.1 (2.15–2.08) | 28.5–1.4 (1.43–1.38) | 50.3–1.5 (1.51–1.46) | 32.1–2.0 (2.03–1.96) | 37.5–1.4 (1.42–1.37) |
| No. reflections | 138,752 (46,479) | 230,405 (26,574) | 29,784 (2950) | 105,501 (10,512) | 235,462 (22,668) | 38,018 (3769) | 171,008 (15,372) |
| $R_{work}/R_{free}$ | 0.21 (0.22)/ 0.26 (0.27) | 0.23 (0.28)/ 0.27 (0.28) | 0.16 (0.19)/ 0.20 (0.25) | 0.16 (0.28)/ 0.19 (0.31) | 0.16 (0.20)/ 0.18 (0.26) | 0.14 (0.16)/ 0.19 (0.22) | 0.16 (0.27)/ 0.19 (0.29) |
| No. atoms | 1203 | 1308 | 2513 | 2985 | 2801 | 2718 | 2999 |
| Protein | 1126 | 1168 | 2324 | 2454 | 2423 | 2263 | 2444 |
| Ligand/ion | 48 | 79 | 53 | 317 | 318 | 309 | 513 |
| Water | 29 | 61 | 136 | 352 | 197 | 279 | 263 |
| *B*-factors | 36.43 | 40.77 | 23.8 | 17.3 | 25.4 | 21.3 | 22.3 |
| Protein | 34.93 | 39.42 | 23 | 15.2 | 24 | 19.7 | 20.7 |
| Ligand/ion | 72.04 | 56.88 | 45.9 | 19.5 | 30.9 | 24.1 | 26.6 |
| Water | 34.56 | 45.86 | 28 | 30.3 | 36.8 | 32.2 | 32.8 |
| R.m.s. deviations | | | | | | | |
| Bond lengths (Å) | 0.009 | 0.012 | 0.003 | 0.011 | 0.009 | 0.005 | 0.016 |
| Bond angles (°) | 1.02 | 2.01 | 0.57 | 1.35 | 1.05 | 0.78 | 1.74 |

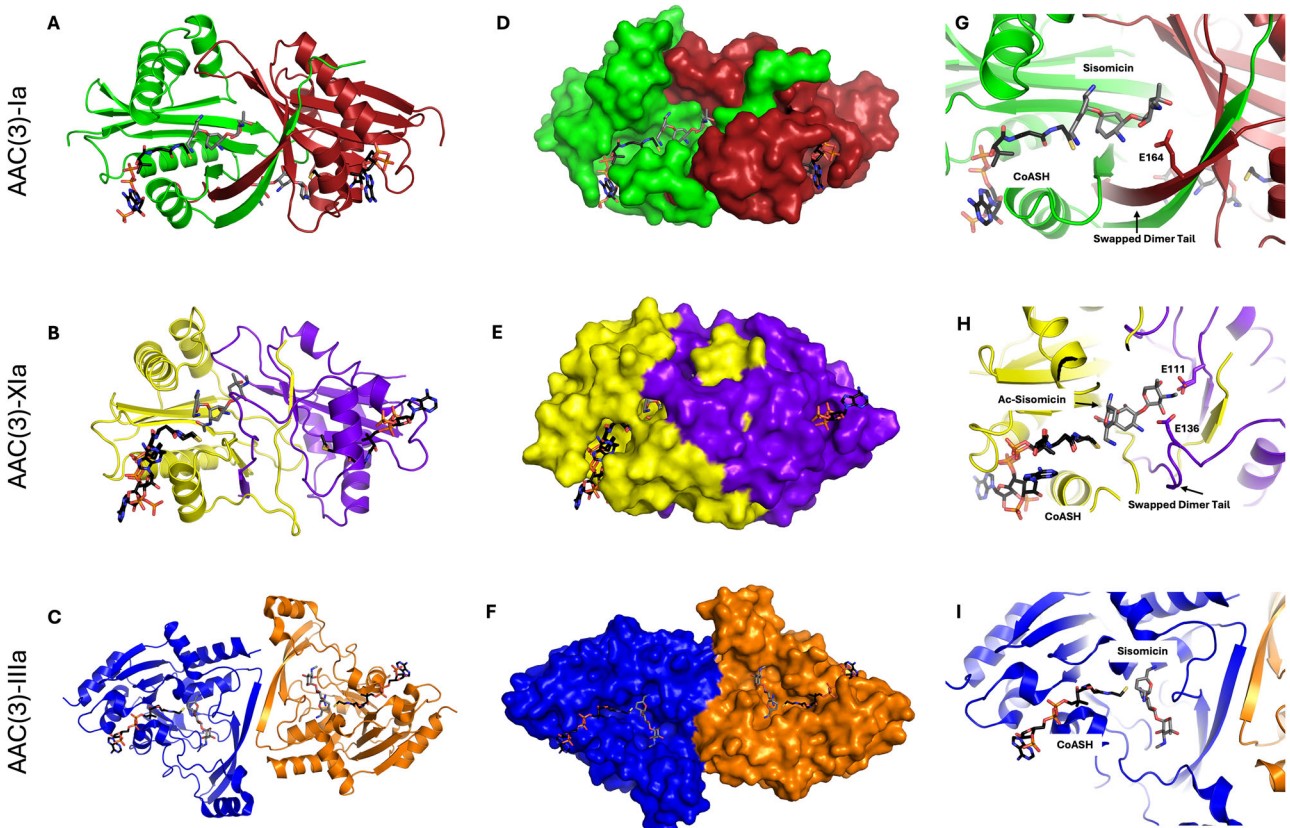

**Fig. 1 | Structural comparison of AAC(3)-Ia, AAC(3)-XIa, and AAC(3)-IIIa.**
**A–C** Global view of AAC(3)-Ia, AAC(3)-XIa, and the previously published AAC(3)-IIIa (PDB: 7MQK) in complex with sisomicin and CoASH. Cartoon representation with dimer subunits colored in green/red for AAC(3)-Ia, yellow/purple for AAC(3)-XIa, and blue/orange for AAC(3)-IIIa. Sisomicin is colored in gray, and CoASH in black, with non-hydrogen atoms colored in the standard red for oxygen, blue for nitrogen, and orange for phosphorus. In the structures of AAC(3)-XIa, two conformations of CoASH are present, and sisomicin has been acetylated. **D–F** Surface representation focused on ligand binding sites. **G–I** Aminoglycoside binding site, focused on the variable region surrounding the double prime ring.

that the overall binding of the CoA molecules is conserved across all of the structures (Fig. 1D, E). In all our structures, the highest concentration of hydrogen bond interactions between enzyme and CoA occurred in the pyrophosphate moiety. The greatest difference between the coenzyme-bound structures of both AAC(3)-Ia and AAC(3)-XIa occurs in the positioning of the adenosine moiety. In the AAC(3)-Ia structures, there is a shift of around 90°, and in the AAC(3)-XIa binary structure with AcCoA, we see a rotation of the ADP moiety of around 180°. The flipping of this group can be attributed to the solvent-exposed nature of the adenine binding pockets of AAC(3)-Ia and AAC(3)-XIa.

The aminoglycoside binding pocket of both AAC(3)-Ia and AAC(3)-XIa are lined with negatively charged residues that form complementary interactions with positively charged aminoglycoside moieties (Supplementary Fig. 2). In the AAC(3)-Ia structure, sisomicin is present in both antibiotic binding sites of the dimer. However, this was not the case for our AAC(3)-XIa structures. Despite a tenfold molar excess of antibiotic to enzyme during crystallization, electron density maps of AAC(3)-XIa only allowed for the positioning of one aminoglycoside per physiological dimer (Fig. 1B). When directly comparing the different protomeric states of AAC(3)-XIa, there are no major structural rearrangements between aminoglycoside-bound and aminoglycoside-lacking complexes.

The shape of the pockets rationalizes these enzymes' preference for 4,6-disubstituted aminoglycosides, as any hexose substituent at O-5 would provoke steric hindrance. In the AAC(3)-Ia and AAC(3)-XIa complexes, regions of the adjacent protomer in the swapped dimer contribute to the aminoglycoside binding sites. These regions are primarily localized around the aminoglycoside double prime ring, with the C-terminal swapped dimer tail located closest to the O-2" substituent (Fig. 1G, H). In the AAC(3)-XIa

structure, the swapped dimer regions are folded closer to the aminoglycoside, creating a more enclosed pocket than is observed in AAC(3)-Ia. In both structures, the C-terminal swapped dimer tail from the adjacent protomer makes direct contact with the aminoglycoside. A glutamic acid residue forms a hydrogen bond with N-3" in AAC(3)-Ia and N-3" in and N-1 in AAC(3)-XIa. In AAC(3)-XIa, there is an additional contact from the adjacent protomer where Glu111 hydrogen bonds with N-3" and O-4" (Fig. 1H)

Overall, the aminoglycoside binding sites of AAC(3)-Ia and AAC(3)-XIa are somewhat similar. They make a comparable number of hydrogen bonds with the aminoglycosides in corresponding positions, but the most notable difference is in the accessibility of the site. In AAC(3)-Ia, the aminoglycoside site is opened and easy for ligands to access (Fig. 1D), whereas in AAC(3)-XIa, the aminoglycoside binding site is closed off (Fig. 1E) and the enzyme would require a conformational shift to accept reactants and release products. While this conformational shift is not seen in any of our crystal structures, the AlphaFold 3 predicted structure of AAC(3)-XIa (Supplementary Fig. 3) had an overall more open conformation. Specifically, the backbone and side chains of the loops spanning residues 30–35 and 136–137 in the tail of the adjacent protomer have shifted. In this position, hydrogen bonds between sisomicin 6′ NH and Asp120$^{O\delta1}$, 1 NH and Asn115$^{O\delta}$, 2″ OH and Glu136$^{O\varepsilon1,2}$, and 2″ OH Asn115$^{N\gamma}$ would be broken. This could possibly represent an intermediate step in the catalytic cycle.

## Non-canonical aminoglycoside binding mode
The experimental maps of the antibiotics in all AAC(3)-Ia and AAC(3)-XIa ternary complex structures presented in this report unambiguously place the aminoglycosides' central ring in a boat conformation rather than the

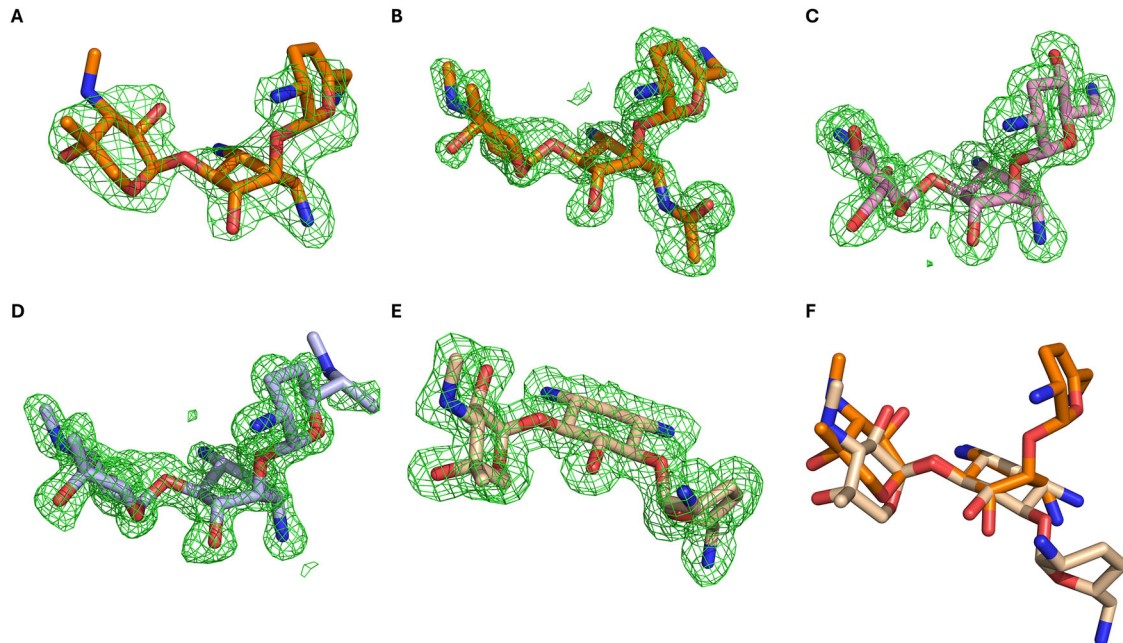

**Fig. 2 | Comparison of sisomicin conformations while bound to resistance enzymes.** Polder maps are shown in green and contoured to 3 σ. **A** Sisomicin boat conformation binding mode from complex with AAC(3)-Ia. **B** acetylated Sisomicin boat conformation binding mode from complex with AAC(3)-XIa. **C** Tobramycin boat conformation binding mode from complex with AAC(3)-XIa. **D** Gentamicin boat conformation binding mode from complex with AAC(3)-XIa. **E** Sisomicin chair conformation binding mode from complex with previously published AAC(3)-IIIa. **F** Overlay for ease of comparison with sisomicin in boat conformation from AAC(3)-Ia complex, colored in orange, and sisomicin in chair conformation from AAC(3)-IIIa complex, colored in wheat.

canonical chair conformation (Fig. 2A–D). Additionally, in all of these aminoglycosides the N-3, O-4 and O-5 atoms of the central ring were positioned in an axial plane rather than an equatorial plane, positioning the prime ring of aminoglycoside around 90° to the central ring (Fig. 2). The uniqueness of this 2-DOS ring conformation was investigated through an analysis of available aminoglycoside-bound macromolecules. All structures in the Protein Data Bank[20] with a resolution of 2.8 Å or higher and containing an aminoglycoside with a real space correlation coefficient (RSCC) of 0.8 or better were geometrically assessed based on dihedral angles of the 2-DOS central ring (Fig. 3). This search covered 219 entries in the PDB and included a total of 442 crystallographically independent aminoglycoside molecules. Of these, none were found to be in a boat configuration. This establishes AAC(3)-Ia and AAC(3)-XIa as the first reported proteins to convincingly stabilize their aminoglycoside substrates in boat conformation.

To evaluate the stability of boat conformation aminoglycosides in solution, density functional calculations were performed on sisomicin crystallographic coordinates that were optimized to estimate the effects of solvation in water. The boat conformation ground state was 7.2 kcal mol$^{-1}$ higher energy than the chair conformation ground state. To estimate the conformational barriers of the central ring, the prime and double prime hexose rings were substituted with methoxy moieties. The boat-chair transition state was found to have a relative energy of 10.3 kcal mol$^{-1}$ greater than the chair conformation ground state. This signifies that boat conformation sisomicin is in an energetic local minimum. The Boltzmann distribution indicates that >99.99% of sisomicin in solution would likely be in chair conformation. However, AAC(3)-Ia and AAC(3)-XIa would likely still have access to boat conformation aminoglycosides, as the interconversion rate between chair and boat conformations was estimated to be $1.3 \times 10^{5}$ s$^{-1}$.

### Comparative structural analysis
To rationalize the ability of AAC(3)-Ia and AAC(3)-XIa to stabilize aminoglycosides in boat conformation, we performed comparative analysis with the prototypical resistance enzyme AAC(3)-IIIa. This enzyme has recently been characterized and was confirmed to stabilize its

aminoglycosides in chair conformation (Fig. 2E)[18]. AAC(3)-IIIa has a high degree of structural similarity to other experimentally determined AAC(3) structures, with RMSD values of 1.7 Å, 1.6 Å, and 1.5 Å with AAC(3)-IIb, AAC(3)-IIIb, and AAC(3)-Xa, respectively (PDB: 7LAO, 6MB4, 7LAP)[21,22]. This makes AAC(3)-IIIa a suitable representative target-mimicking aminoglycoside resistance enzyme for our analyses.

Overall, AAC(3)-Ia and AAC(3)-XIa are both notably smaller than AAC(3)-IIIa, with monomeric molecular masses of 16.9 kDa and 16.4 kDa, respectively, for AAC(3)-Ia and AAC(3)-XIa, while AAC(3)-IIIa is 29.9 kDa. There are also differences within the aminoglycoside binding sites. Both known boat-binding resistance enzymes use the swapped-dimer C-terminal tail from the other monomer as part of their aminoglycoside binding site, while AAC(3)-IIIa did not use swapped-dimer tails for stabilizing ligands (Fig. 1). Overall, the boat-binding resistance enzymes stabilized their aminoglycosides in a more extensive manner, making contacts with all three rings. Whereas AAC(3)-IIIa only made contacts with the central ring and the double prime ring, leaving the prime ring solvent exposed. In total, the boat-binder AAC(3)-Ia made 16 hydrogen bonds with sisomicin, and AAC(3)-XIa made 17 hydrogen bonds with acetylated sisomicin, while AAC(3)-IIIa only made 7 hydrogen bonds with sisomicin (Fig. 4 and Supplementary Fig. 4).

### Kinetics and affinity of aminoglycoside binding
Michaelis-Menten kinetics assays were performed to compare the catalytic efficiencies of the boat-binding resistance enzymes AAC(3)-Ia and AAC(3)-XIa against the chair-binding enzyme AAC(3)-IIIa (Table 2). These experiments found that AAC(3)-IIIa was significantly more catalytically efficient ($P \leq 0.01$) than either AAC(3)-Ia or AAC(3)-XIa in reactions with sisomicin or gentamicin (Fig. 5). AAC(3)-IIIa had a $k_{cat}/K_M$ of 2.5 ± 0.8 and 2.7 ± 1.1 µM$^{-1}$·s$^{-1}$ for sisomicin and gentamicin, respectively. AAC(3)-Ia had a $k_{cat}/K_M$ of 0.06 ± 0.01 and 0.05 ± 0.01 µM$^{-1}$·s$^{-1}$ for sisomicin and gentamicin, respectively. AAC(3)-XIa had a $k_{cat}/K_M$ of 0.06 ± 0.01 and 0.2 ± 0.04 µM$^{-1}$·s$^{-1}$ for sisomicin and gentamicin, respectively.

Isothermal titration calorimetry (ITC) was performed to examine the thermodynamic aspects of target mimicry, comparing the non-target-

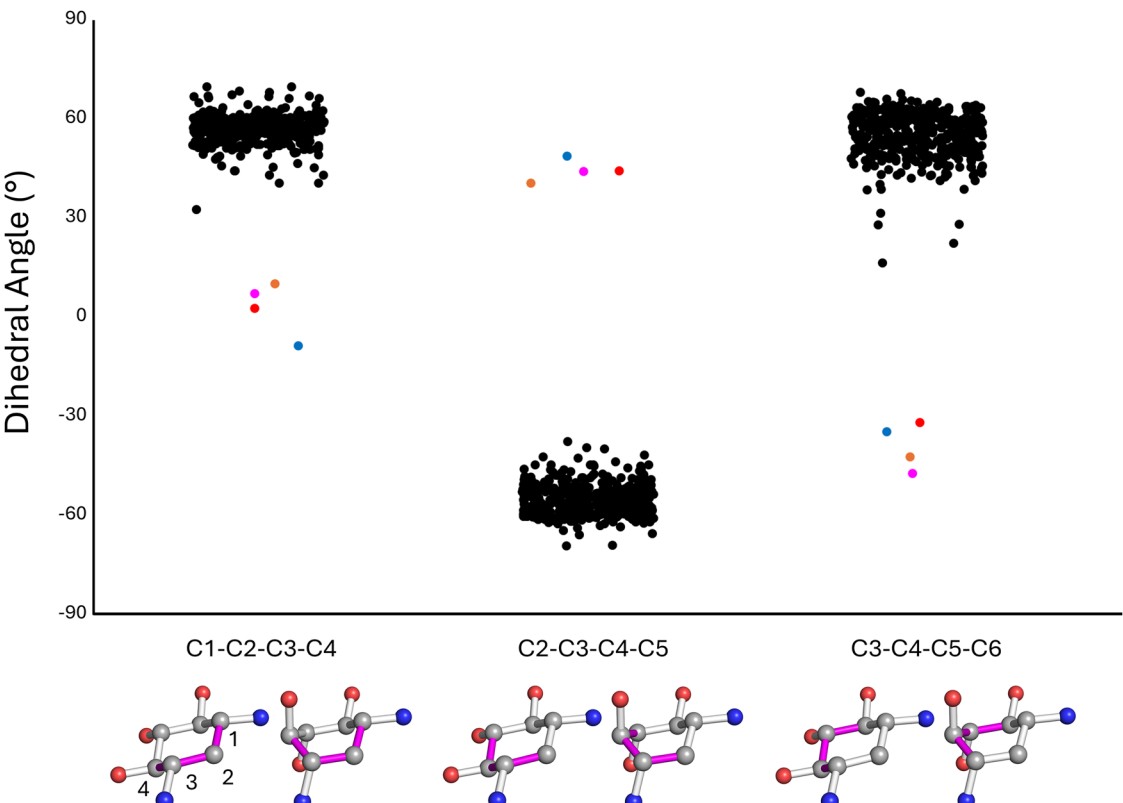

**Fig. 3 | Dihedral angle analysis of Protein Data Bank.** Jitter plot of dihedral angles in the 2-DOS ring of all aminoglycoside-containing structures in the Protein Data Bank that have a resolution of ≥2.8 Å, and a real-space correlation coefficient of ≥0.8, as of submission date (January 21, 2025). This analysis included 219 PDB entries and a total of 442 crystallographically independent aminoglycoside molecules. Black dots represent previously deposited structures, and colored dots represent structures presented in this report, with magenta as AAC(3)-Ia bound sisomicin, orange as AAC(3)-XIa bound acetylated sisomicin, red as AAC(3)-XIa bound gentamicin, and blue as AAC(3)-XIa bound tobramycin. The only four aminoglycoside molecules that displayed dihedral angles associated with boat conformation were those associated with the structures of AAC(3)-Ia and AAC(3)-XIa presented in this work. This establishes AAC(3)-Ia and AAC(3)-XIa as the first identified enzymes that convincingly bind to aminoglycosides with their central 2-DOS ring stabilized in boat conformation.

mimicking enzymes AAC(3)-Ia and AAC(3)-XIa against the target-mimicking enzyme AAC(3)-IIIa. ITC experiments were conducted using coenzyme A in combination with sisomicin or gentamicin as aminoglycoside substrates. Thermodynamic binding parameters are presented in Table 3, and representative ITC profiles are provided (Supplementary Fig. 5). The ITC curves were best described by a two-set of sites model. Overall, AAC(3)-Ia ($K_{D\ Sisomicin}$ = 26.0 ± 0.6 μM, $K_{D\ Gentamicin}$ = 17.1 ± 3.2 μM) had a significantly lower affinity than AAC(3)-IIIa ($K_{D\ Sisomicin}$ = 1.3 ± 0.1 μM, $K_{D\ Gentamicin}$ = 8.3 ± 0.2 μM) and AAC(3)-XIa ($K_{D\ Sisomicin}$ = 1.1 ± 0.02 μM, $K_{D\ Gentamicin}$ = 6.1 ± 1.8 μM) for both sisomicin and gentamicin. There was no significant difference in affinity between AAC(3)-IIIa and AAC(3)-XIa for either sisomicin or gentamicin (Fig.6).

**Antibiotic susceptibility testing**

Antibiotic susceptibility testing was performed to assess the implications of non-canonical aminoglycoside binding on the ability of resistance enzymes to compete with the ribosome, and thus confer in vivo antibiotic resistance. First, RT-qPCR analysis was performed on *E. coli* cultures (*n* = 3 experimental replicants) to ensure that antibiotic susceptibility testing results were not impacted by differential expression levels. Experiments measured fold change against the reference gene *hcaT*, as recommended in ref. 23. RT-qPCR experiments were analyzed with a one-way ANOVA, and there was no significant difference in transcription levels of genes for AAC(3)-Ia, AAC(3)-IIIa, or AAC(3)-XIa (Supplementary Fig. 6). Note that efforts to assess protein levels by Western blot analysis were also performed, but these experiments only revealed that this technique lacks the sensitivity required (see Supplementary Data). Growth curves (*n* = 4 technical

replicants) were conducted in BL21(DE3) *E. coli* transformed with pUCP18 vectors expressing untagged AAC(3)-Ia, AAC(3)-XIa, and AAC(3)-IIIa. OD600 values from the empty vector control were subtracted from experimental values to isolate the effects of the resistance gene inserts (Fig. 7A). At a concentration of 12 μg mL$^{-1}$ of sisomicin or gentamicin, none of the strains of transformed *E. coli* were able to grow at all. AAC(3)-Ia was unable to outperform the empty vector control in any experiments. Both AAC(3)-XIa and AAC(3)-IIIa transformed cultures provided a greater level of resistance than the empty vector control against 6 μg mL$^{-1}$ sisomicin and 3 μg mL$^{-1}$ gentamicin. AAC(3)-XIa provided greater resistance to sisomicin than AAC(3)-IIIa, but both AAC(3)-XIa and AAC(3)-IIIa provided similar levels of resistance to gentamicin. MIC assays were performed on BL21(DE3) *E. coli* transformed with pUCP18 vectors expressing AAC(3)-Ia, AAC(3)-XIa, and AAC(3)-IIIa. These assays were conducted on N-terminal 6xHis-tagged fusion-proteins, as well as untagged wild-type proteins. No difference was observed between these two groups. Assays had n = 3 experimental replicants with *n* = 3 technical replicants each and included sisomicin and gentamicin. To eliminate any effects that the pUCP18 vector may have on antibiotic susceptibility, control experiments were also performed with an empty pUCP18 vector. The minimum inhibitory concentration of sisomicin for AAC(3)-Ia, AAC(3)-Xia, AAC(3)-IIIa, and empty pUCP18 vector transformed *E. coli* were 6, 12, 12, and 6 μg mL$^{-1}$, respectively (Fig. 7 B). The minimum inhibitory concentration of gentamicin for AAC(3)-Ia, AAC(3)-Xia, AAC(3)-IIIa, and empty pUCP18 vector transformed *E. coli* were the same as sisomicin, also with 6, 12, 12, and 6 μg mL$^{-1}$, respectively (Fig. 7 B). Results from both the tagged and untagged MIC experiments were in complete agreement with the growth curve data.

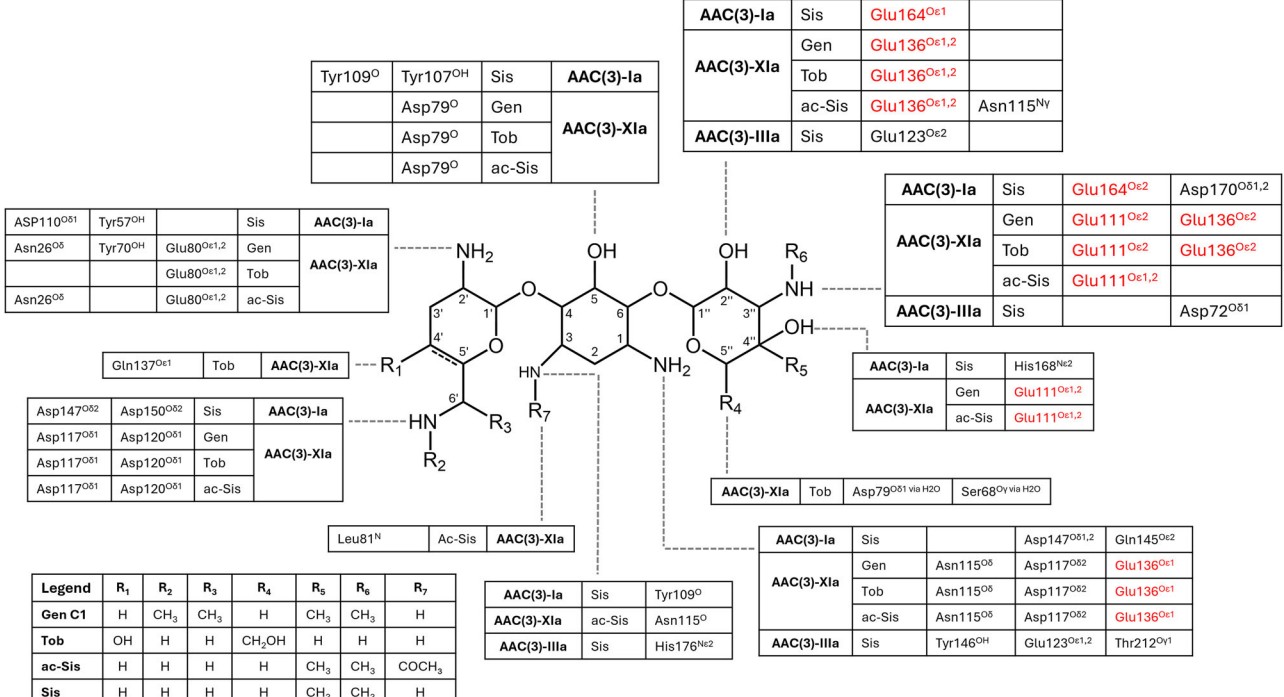

**Fig. 4 | Protein–ligand interactions between aminoglycosides and resistance enzymes.** Combined structures of gentamicin C1 (Gen), tobramycin (Tob), acetylated sisomicin (ac-Sis), and unacetylated sisomicin (Sis). Tables linked to aminoglycoside atoms indicate hydrogen bond interactions. Atoms with font colored in red indicate atoms from the adjacent swapped dimer subunit.

## Discussion

AACs are a major branch of enzymes responsible for enzyme-mediated aminoglycoside resistance[5,14,24]. Currently, this family comprises nearly 70 known isozymes, representing 14 enzymes with unique substrate specificities[5,14,24]. AAC(3)-Ia was the first plasmid-encoded acetyltransferase identified from clinical isolates of *P. aeruginosa* in 1972[11–13]. Recently, a new aminoglycoside-modifying enzyme, AAC(3)-XIa, was discovered in the bacterial pathogen *C. striatum*. No structures of either of these enzymes in complex with aminoglycosides had previously been reported. In this work, we present two structures of AAC(3)-Ia and five structures of AAC(3)-XIa in various states.

Aminoglycosides bound to macromolecules with central rings stabilized in boat conformation are extremely rare. Our analysis of all aminoglycosides bound to macromolecules in the PDB found no previously deposited structures containing aminoglycosides convincingly modeled in boat conformation (Fig. 3). While our density functional calculations estimate that >99.99% of sisomicin in solution will be in chair conformation, the fast interconversion rate between chair and boat conformations of $1.3 \times 10^5 \, \text{s}^{-1}$ indicates that resistance enzymes would have access to boat conformation aminoglycosides. In boat conformation of 4,6-disubstituted aminoglycosides, the prime ring is in an axial position, rather than the more stable equatorial position observed in chair conformation (Fig. 2). This 90°

rotation has a large impact on how these aminoglycosides interact with resistance enzymes. When chair conformation aminoglycosides are modeled into the binding site of AAC(3)-Ia or AAC(3)-XIa, steric clashes between the aminoglycoside prime ring and beta sheet backbone atoms occur (Supplementary Fig. 7). When the conformation of aminoglycosides are modeled into the binding pocket of AAC(3)-IIIa, steric clashes occur between the aminoglycoside prime ring and the side chains of residues lining the active site. These results indicate that these enzymes would be unable to bind non-preferred conformation aminoglycosides. When binding their favored conformation, we observed that the two boat-binding resistance enzymes made 9–10 more hydrogen bonds with sisomicin than the chair-binding resistance enzyme AAC(3)-IIIa (Fig. 4 and Supplementary Fig. 4). Furthermore, the boat-binding resistance enzymes both make 4 hydrogen bonds each with the prime sisomicin ring, whereas AAC(3)-IIIa did not make any hydrogen bonds with the solvent-exposed aminoglycoside prime ring, emphasizing the active site architectural differences between these boat- and chair-binding resistance enzymes.

To measure the performance of non-target-mimicking, boat-binding resistance enzymes, catalytic efficiency, binding affinity, and bacterial antibiotic susceptibility were assessed. Experiments were performed on the boat-binding resistance enzymes AAC(3)-Ia and AAC(3)-XIa, as well as the prototypical chair-binding resistance enzyme AAC(3)-IIIa. Throughout these experiments, sisomicin and gentamicin were used, as they are substrates for all three enzymes. Enzymatic assays found that AAC(3)-IIIa had a significantly higher ($P \le 0.01$) catalytic efficiency than either of the boat-binding enzymes with either sisomicin or gentamicin substrates (Fig. 5). This supported the position that target-mimicking resistance enzymes may be higher performing than non-target mimicking. However, further analysis of boat-binder performance did not align as well. To assess the potential thermodynamic effects of binding boat conformation aminoglycosides, isothermal titration calorimetry was performed. ITC curves were fit with a two-set of sites model, where each site was filled to approximately half occupancy. This suggests that each protomer has a single site that binds aminoglycosides with two sets of energetic binding conditions. While this may suggest that these AACs are able to bind both boat and chair

## Table 2 | Enzymatic assay results

| Enzyme | Aminoglycoside | kcat (s$^{-1}$) | KM (µM) | kcat/KM (µM$^{-1}$·s$^{-1}$) |
|---|---|---|---|---|
| AAC(3)-Ia | Sisomicin | 3.1 ± 0.2 | 53 ± 6 | 0.06 ± 0.01 |
| | Gentamicin | 2.0 ± 0.1 | 37 ± 5 | 0.05 ± 0.01 |
| AAC(3)-XIa | Sisomicin | 0.4 ± 0.01 | 6.8 ± 0.7 | 0.06 ± 0.01 |
| | Gentamicin | 0.2 ± 0.003 | 0.8 ± 0.2 | 0.2 ± 0.04 |
| AAC(3)-IIIa | Sisomicin | 45 ± 5 | 18 ± 4 | 2.5 ± 0.8 |
| | Gentamicin | 36 ± 5 | 13 ± 4 | 2.7 ± 1.1 |

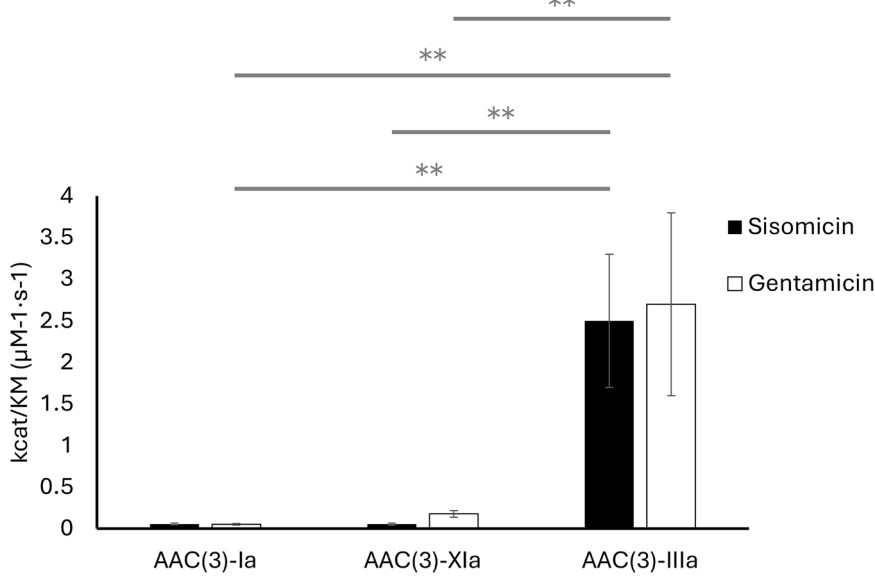

**Fig. 5 | Catalytic efficiency of resistance enzymes with sisomicin and gentamicin.** Error bars depict the standard error of the mean. $n = 3$, analyzed using a one-way ANOVA with a Tukey multiple comparison test. **$P \leq 0.01$.

conformation aminoglycosides, our structural data disagrees. As discussed above, due to steric clashes, the resistance enzymes assessed were unable to bind non-preferred aminoglycoside conformers (Supplementary Fig. 7). Further studies are required to determine the cause of these two sets of energetic binding conditions. Ultimately, the ITC experiments found that AAC(3)-Ia had a significantly lower affinity than AAC(3)-XIa and AAC(3)-IIIa for both sisomicin and gentamicin, while no significant difference in binding affinity was observed between AAC(3)-XIa and AAC(3)-IIIa (Fig. 6). Overall, each enzyme had thermodynamically favorable free energy for binding, although enthalpy and entropy contributions varied with each enzyme. When target mimicry was first proposed, it was postulated that target mimicry may increase affinity through a reduction in the loss of entropy[6]. While our results for the unfavorable entropy of boat-binder AAC(3)-XIa may support this notion, boat-binder AAC(3)-Ia and chair-binder AAC(3)-IIIa presented a similar range of entropies, suggesting that aminoglycoside conformation preference may not have a consequential impact on the binding entropy. The overall flexibility of aminoglycosides in solution is likely a greater entropic contributor than the specific aminoglycoside conformation in complex with enzyme[25].

Antibiotic susceptibility testing was used to assess the effects of boat-binding resistance enzymes in vivo. Susceptibility testing was performed as growth curves and end-point minimum inhibitory concentration (MIC) experiments. RT-qPCR experiments found that transformed E. coli strains expressed genes encoding resistance enzymes without any significant variation (Supplementary Fig. 6). In susceptibility growth curves, AAC(3)-XIa appeared to provide the highest level of resistance against sisomicin and a similar level of resistance against gentamicin as AAC(3)-IIIa. However, both AAC(3)-XIa and AAC(3)-IIIa had the same MIC value for both antibiotics. It was expected that AAC(3)-XIa and AAC(3)-IIIa would provide similar levels of resistance against both antibiotics, as they did not have significantly different affinities for these substrates in ITC experiments. In susceptibility tests for AAC(3)-Ia against gentamicin and sisomicin, AAC(3)-Ia was susceptible to both antibiotics and had the same MIC values as the empty vector control. While this was unexpected for a resistance factor that is present in clinical isolates, it is understood that our experiments do not perfectly recapitulate real-world conditions where this enzyme may provide better resistance.

Integration of the susceptibility testing data with data from enzymatic assays, ITC, and structural studies highlighted the catalytic efficiency of AAC(3)-XIa. This enzyme has a similar catalytic efficiency to AAC(3)-Ia and a lower catalytic efficiency than AAC(3)-IIIa, yet can provide similar antibiotic resistance to AAC(3)-IIIa. The low catalytic efficiency of the boat-binding enzymes is likely due to the number of hydrogen bonds required to stabilize the aminoglycoside in this atypical conformation. Since a greater number of hydrogen bonds must be formed and broken per catalytic cycle, this could slow the rate of acetylation. Despite having a low catalytic efficiency, the level of antibiotic resistance provided by AAC(3)-XIa may be due to its higher affinity for aminoglycosides. In ITC experiments, AAC(3)-XIa displayed a significantly higher affinity for both sisomicin and gentamicin than AAC(3)-Ia. A potential rationalization for this could be that AAC(3)-Xia's higher affinity paired with low turnover of ligands is allowing it to sequester enough antibiotic to offset the effects of low catalytic efficiency. However, Aminoglycoside modifying enzymes providing resistance through sequestration is exceedingly rare, only documented twice[26,27].

## Conclusions

The near universal presence of target mimicry in enzyme mediated aminoglycoside resistance, indicates a strong evolutionary pressure for exploiting the lowest energy conformation of the antibiotic in binding to these resistance enzymes. The unprecedented observation that two enzymes defy the use of target mimicry questions our initial hypothesis that target mimicry is necessary to effectively compete with the antibiotic's natural target, the ribosome. While the data for AAC(3)-Ia appears to support the importance of target-mimicry, AAC(3)-XIa exemplifies that non-target-mimicking resistance factors can have similar affinities to target-mimicking resistance factors and can confer effective resistance against aminoglycoside antibiotics. Nonetheless, the frequency with which this is observed suggests that target mimicry is a preferred strategy in the aminoglycoside resistome.

## Materials and methods
### Gene synthesis
The AAC(3)-Ia gene from *A. baumannii* (NCBI: WP_063840259.1), AAC(3)-IIIa gene from *P. aeruginosa* (GenBank: CAA39184.1), and AAC(3)-XIa gene from *C. striatum* (NCBI: NG_050595) were synthesized by BioBasic Inc. with codons optimized for expression in *E. coli*, in pET15b expression vectors between the NdeI and BamHI restriction sites, resulting in a 6xHis-tag that could be removed through thrombin cleavage. The resulting vectors were used to transform LOBSTR *E. coli* cells[28]. For antibiotic susceptibility testing, three sets of the above genes were synthesized by BioBasic Inc. using the same DNA sequences that were used for the pET-15b constructs, in pUCP18 expression vectors between the BamHI and EcoRI restriction sites. The first set did not include any type of fusion tag, the second set contained a N-terminal 6×His-tag with a glycine-serine-glycine linker, and the third set contained a N-terminal FLAG-tag with a

**Table 3 | Isothermal titration calorimetry results**

| Aminoglycoside | AAC(3)-Ia | | AAC(3)-XIa | | AAC(3)-IIIa | |
| --- | --- | --- | --- | --- | --- | --- |
| | Sisomicin | Gentamicin | Sisomicin | Gentamicin | Sisomicin | Gentamicin |
| N (sites) | 0.50 ± 0.06 | 0.49 ± 0.02 | 0.51 ± 0.01 | 0.28 ± 0.06 | 0.35 ± 0.03 | 0.63 ± 0.01 |
| Kd (uM) | 26.0 ± 0.6 | 17.1 ± 3.2 | 1.1 ± 0.02 | 6.1 ± 1.8 | 1.3 ± 0.1 | 8.3 ± 0.2 |
| ΔH (cal/mol) | −2950 ± 416 | −7330 ± 238 | −12,600 ± 372 | −6770 ± 2870 | −8460 ± 351 | −3340 ± 56 |
| TΔS (cal/mol) | 3300 ± 427 | −808 ± 343 | −4510 ± 378 | −1380 ± 171 | −411 ± 318 | 3600 ± 53 |
| ΔG (cal/mol) | −6250 ± 596 | -6520 ± 418 | −8120 ± 530 | −5380 ± 2880 | −8050 ± 474 | −6940 ± 77 |
| N2 (sites) | 0.54 ± 0.01 | 0.57 ± 0.02 | 0.36 ± 0.04 | 0.31 ± 0.05 | 0.56 ± 0.04 | 0.63 ± 0.01 |
| Kd2 (uM) | 71.8 ± 7.5 | 23.8 ± 0.3 | 27.1 ± 4.2 | 8.5 ± 5.5 | 2.1 ± 0.2 | 12.4 ± 0.5 |
| ΔH 2 (cal/mol) | −2210 ± 763 | −2060 ± 80 | 1058 ± 1548 | −9958 ± 567 | −5740 ± 143 | −1460 ± 63 |
| TΔS 2 (cal/mol) | 3450 ± 795 | 4240 ± 80 | 7295 ± 1615 | −2916 ± 1036 | 2000 ± 146 | 5240 ± 43 |
| ΔG 2 (cal/mol) | −5660 ± 1100 | −6300 ± 112 | −6237 ± 2237 | −7042 ± 1180 | −7740 ± 204 | −6690 ± 77 |

Thermodynamic parameters obtained from calorimetric titration of AAC(3)-XIa, AAC(3)-Ia, and AAC(3)-IIIa with sisomicin and gentamicin, $n = 3$.

glycine-serine-serine-glycine linker. The resulting vectors were used to transform BL21(DE3) *E. coli* cells.

## Protein expression
All protein expression was performed according to the Studier method for autoinduction[29]. A 1 mL noninducing starter culture grown for 4 h at 37 °C and 260 was used to inoculate 1 L of autoinducing media containing 100 mg mL⁻¹ ampicillin. The culture was incubated at 37 °C for 3 h, followed by 20 °C for 16 h at 220 rpm.

## Protein purification
As per previous publication[18], except AAC(3)-Ia storage buffer contained 20 mM HEPES pH 7.5, 150 mM NaCl. AAC(3)-XIa used the same storage buffer as AAC(3)-IIIa, as discussed in previous work.

## Synthesis of Ac$_{NH}$CoA
Amino-pantetheine (WuXi AppTec) was converted to amino-CoA using a previously reported one-pot biochemical reaction[30]. Amino-CoA was subsequently added with acetyl *N*-hydroxy succinimide (Zamboni Chemical Solutions) and *N,N*-diisopropylethylamine (Sigma-Aldrich) in a 1:8:4 molar ratio, respectively, in 80% *N,N*-dimethylformamide (Sigma-Aldrich)[31]. The coupling reaction was stirred overnight at room temperature. Based on previous protocols, the acetyl amino-CoA was purified by preparatory high-performance liquid chromatography (HPLC) and verified by high-resolution mass spectrometry[31]. The purity was assessed by analytical HPLC.

## AAC(3)-Ia crystallization
Crystals of AAC(3)-Ia were grown at 22 °C using the sitting-drop vapor diffusion method and optimized using the microseeding. The initial crystals used for seed stock preparation were obtained from co-crystallization of 10 mg mL⁻¹ AAC(3)-Ia with a 10× molar excess of sisomicin and CoASH in 0.1 M Tris, pH 8.5, 0.05 M MgCl2, 40% EtOH. The final crystals were obtained in 0.1 M Tris Ph 8.5, 0.05 M MgCl2, 20% EtOH, 35% PEG 400, co-crystallized with a 10× molar excess of gentamycin and CoASH. However, discovery maps from crystals obtained in this condition only had density for CoASH, and not gentamicin, as seen in the presented AAC(3)-Ia and CoASH structure. For the structure of AAC(3)-Ia in complex with CoASH and sisomicin, the crystal was soaked for about 3 min in crystallization solution supplemented with 10 mM sisomicin before freezing for data collection.

## AAC(3)-XIa crystallization
Crystals of AAC(3)-XIa were also grown at 22 °C using the sitting-drop vapor diffusion method and optimized using the microseeding. The initial crystals used for seed stock preparation were obtained when drops contained a 1:1 ratio of 10 mg mL⁻¹ AAC(3)-XIa in storage buffer supplemented with 3 mM CoASH and 6 mM aminoglycoside, and reservoir solution that consisted of 0.1 M trisodium citrate (pH 4.0), 40% (*v/v*) ( ± )-2-methyl-2,4-pentanediol (MPD) and 3.3% (v/v) PEG200. Final crystals of AAC(3)-XIa grew when drops contained a 3:2:1 ratio of 10 mg mL⁻¹ protein solution: reservoir solution: diluted seed stock, where the reservoir solution consisted of 0.1 M trisodium acetate, pH 4.6, 10–30% (*v/v*) MPD, and 0–24% PEG 200.

## Data collection and structure determination
Diffraction data for the AAC(3)-Ia complex with sisomicin and CoASH were collected on the Canadian Light Source CMCF-BM beamline coupled with a PILATUS 6 M detector. The data was indexed, integrated, and scaled in HKL2000[32]. This structure was solved by molecular replacement (MR) using PHASER[33], with AAC(3)-Ia in complex with CoASH (PDB: 6BVC), stripped of all non-protein atoms as search model. Diffraction data for the structure of AAC(3)-Ia in complex with CoASH were collected on the Canadian Light Source CMCF-ID beamline coupled with an Eiger X 9 M detector. The raw data were processed in DIALS[34]. This structure was solved by difference Fourier synthesis in phenix.refine[35] using the previous structure of AAC(3)-Ia in complex with sisomicin and CoASH. Diffraction data for AAC(3)-XIa partial-apo structure and in complex with AcCoA and tobramycin + CoASH were collected on a Bruker D8 Venture home source consisting of a METALJET X-ray source (liquid gallium) coupled with a PHOTON II CPAD detector mounted on a KAPPA goniometer. Datasets for those structures were processed using the PROTEUM software suite (version 2018.7-2). AAC(3)-XIa in complex with CoASH and tobramycin was solved by MR using PHASER[33], with an AAC(3) homolog (PDB ID: 2R1I), stripped of all non-protein atoms as the search model (version 2.8.3). Diffraction data for AAC(3)-XIa in complex with Ac$_{NH}$CoA were obtained at Advanced Light Source beamline 5.0.1. Dataset for this structure was processed using xia2 pipeline [DIALS][36]. All subsequent AAC(3)-XIa structures were solved using AAC(3)-XIa stripped of ligands as a search model for molecular replacement. All structures were refined by iterative cycles of real-space refinement and model building in Coot[37] and reciprocal-space refinement with either phenix.refine[35] for the AAC(3)-Ia structure with CoASH, and all AAC(3)-XIa structures, or REFMAC[38] for the AAC(3)-Ia complex with sisomicin and CoASH. The AAC(3)-Ia structure with sisomicin + CoASH suffered from ice rings present throughout the dataset, which resulted in lower R-factors than expected. Structural models were deposited in the PDB (PDB IDs: 9MGS, 9MGU, 9MGT, 9MH5, 9MH3, 9MH6, and 9MH7). The final data collection and refinement statistics are summarized in Table 1.

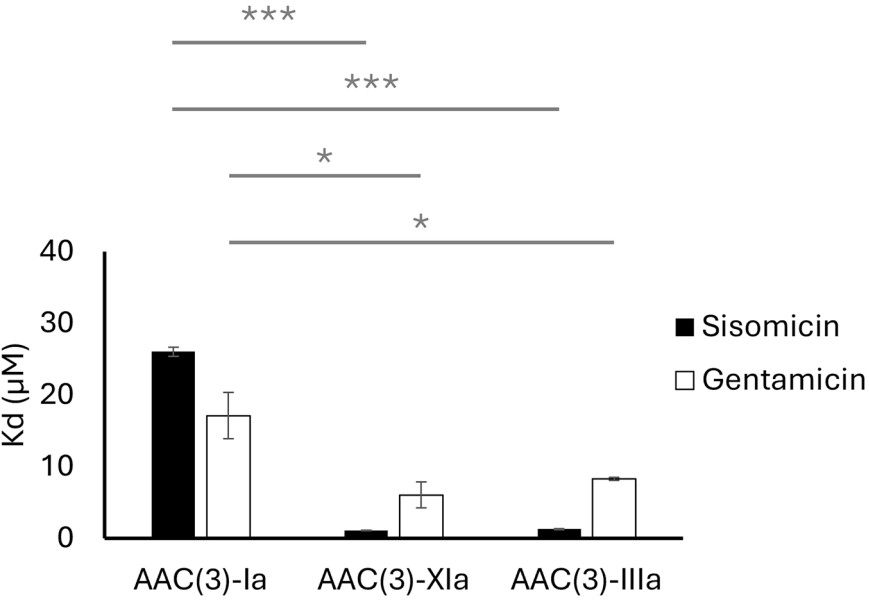

**Fig. 6 | Affinity of aminoglycoside resistance enzymes for sisomicin and gentamicin.** Measured through ($n = 3$) isothermal titration calorimetry experiments. Error bars depict the standard error of the mean. Analyzed using a one-way ANOVA with a Tukey multiple comparison test. $^{*}P \le 0.05$, $^{***}P \le 0.001$.

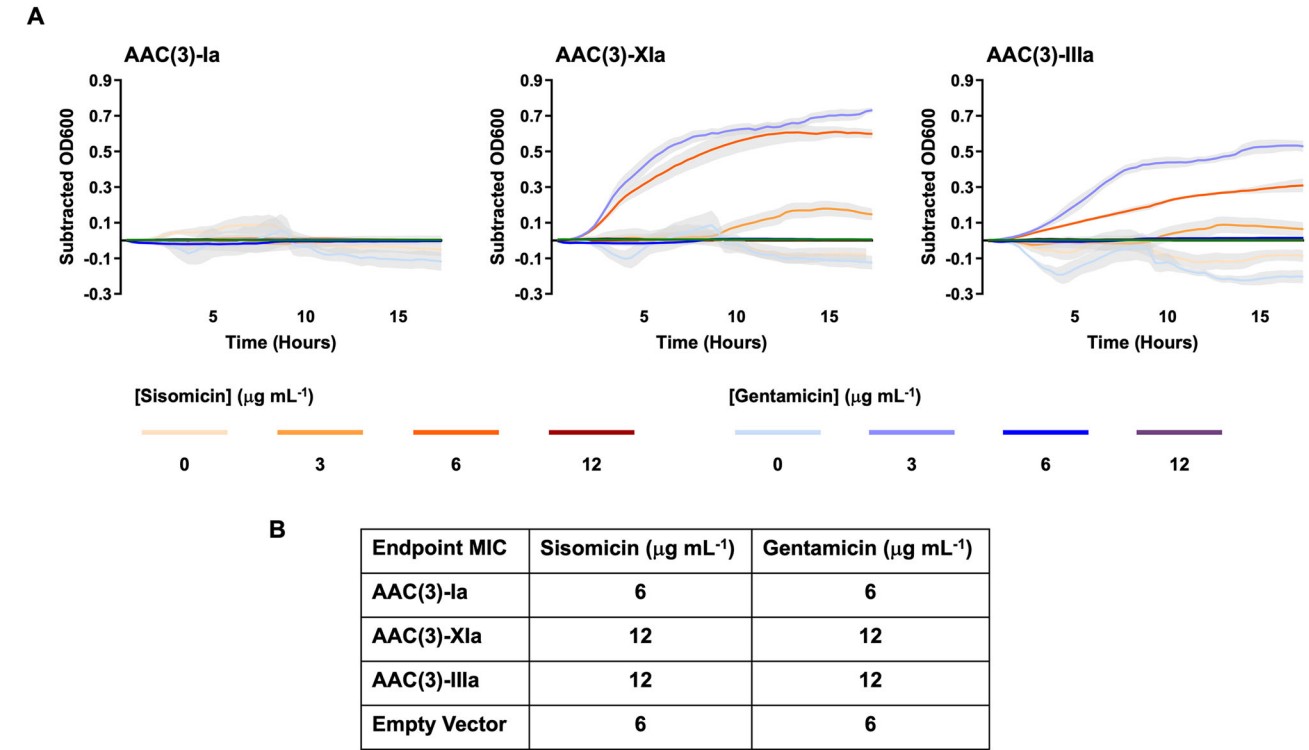

| Endpoint MIC | Sisomicin (µg mL⁻¹) | Gentamicin (µg mL⁻¹) |
|---|---|---|
| AAC(3)-Ia | 6 | 6 |
| AAC(3)-XIa | 12 | 12 |
| AAC(3)-IIIa | 12 | 12 |
| Empty Vector | 6 | 6 |

**Fig. 7 | Aminoglycoside susceptibility testing of *E. coli* expressing resistance genes.** *E. coli* were transformed with pUCP18 plasmids carrying genes for AAC(3)-Ia, AAC(3)-IIIa, and AAC(3)-XIa and exposed to increasing concentrations of sisomicin and gentamicin. **A** Growth curves of *E. coli* expressing untagged wild-type resistance genes. OD600 measured on 20-min intervals over 17 h. Growth curves depict OD600 values after empty vector subtraction to visualize the impact of each resistance gene insert ($n = 4$ technical replicants). Higher culture turbidity indicates a higher degree of resistance to the assessed antibiotic. Error envelopes shown in light gray depict the standard error of the mean. Media control shown in green. **B** Endpoint minimum inhibitory concentration (MIC) results of *E. coli* expressing resistance genes. Measured after 20 h of growth. Same results found with and without N-terminal 6×His fusion tags ($n = 3$ experimental replicants with $n = 3$ technical replicants each).

## Density functional calculations

Density functional calculations were performed with B3LYP functionals[39] and Dunnings correlation consistent cc-pvdz basis set using Gaussian16[40,41] implemented on Compute Canada Graham cluster. Initial coordinates for the heavy atoms of sisomicin were imported from crystallographic structures, and hydrogens were placed in calculated positions using Gauss View 6.1. These preliminary structures were then optimized as gas-phase geometries with configurations allowing for internal hydrogen bonding. After optimization, the polarized continuum model (PCM) was used to estimate the effect of solvation in water. All structures correspond to local minima as determined by post-optimization frequency determinations. To estimate the conformational barriers present in the cyclohexane middle ring of

sisomicin, a dimethoxy substitution was used with chair and boat conformations optimized with B3LYP functionals and a cc-pvtz basis set. The transition state connecting these two local minima was then determined from frequency calculations on the identified stationary states. Density functional calculation energies of optimized conformers can be found in supplementary Table 1, and Sisomicin cartesian coordinates used for density functional theory calculations, as optimized by DFT calculations, can be found in Supplementary Table 2.

## Enzymatic assay

The kinetic data of AAC(3)-Ia, AAC(3)-XIa, and AAC(3)-IIIa against gentamicin and sisomicin were obtained using the ThermoFisher NanoDrop OneC Spectrometer. The acetylation of aminoglycosides was measured by a coupled assay, where the formation of pyridine-4-thiolate can be detected at 324 nm[42,43]. The kinetic assays were performed at room temperature (22 °C) in quartz cuvettes (pathlength 1 cm) at a final volume of 0.8 mL. Reaction solution contained 25 mM MOPS pH 6.5, 100 mM NaCl, 500 μM 4,4'-dipyridyl disulfide (Aldrithiol™-4, Sigma-Aldrich), 150 μM acetyl coenzyme A, and aminoglycoside concentrations varying from 1.25 to 160 μM. The reaction was initialized by the addition of AAC(3)-Ia to a final concentration of 0.2 μM, AAC(3)-XIa to a final concentration of 1.94 μM, or AAC(3)-IIIa to a final concentration of 0.4 μM, where UV absorbance was measured over 5 min. All reactions were run in triplicate, and data analysis was performed using GraphPad5 software. One-way ANOVA and Tukey's multiple comparison tests were used to determine the statistical significance of catalytic efficiencies.

## Isothermal titration calorimetry

To accurately measure protein ligand interactions, the concentration of our proteins of interest was measured spectrophotometrically using the NanoDrop OneC Spectrometer (Thermo Scientific). ITC experiments were performed on a MicroCal iTC 200 (General Electric) at 25 °C with stirring at 750 rpm. Unique buffer solutions were used for each enzyme. AAC(3)-Ia was prepared in 50 mM HEPES, pH 7, and 150 mM NaCl. AAC(3)-IIIa was prepared in 120 mM Tris, pH 7.5, and 150 mM NaCl. AAC(3)-XIa was prepared in 20 mM Tris, pH 7.5, and 150 mM NaCl. The cell contained 550 μM AAC(3)-Ia, 350 μM AAC(3)-IIIa, or 450 μM AAC(3)-XIa. All experiments also contained 5000 μM of coenzyme A trilithium salt in the cell. For experiments with AAC(3)-Ia, AAC(3)-IIIa, and AAC(3)-XIa, the syringe was loaded with 12000 μM and 3150 μM of sisomicin or gentamicin, respectively, with the latter two enzymes using the same aminoglycoside concentrations. In all ITC experiments, aminoglycosides were prepared in the same buffering solution as the target protein. The ligands were injected 29 times (21.3 μl per injection) with a 180 s interval between injections. Results were analyzed using ORIGIN software version 7E (MicroCal) using the molecular weight of a single protomer of each protein and fitted with a two-set of sites binding model (Supplementary Fig. 5). Data analysis was performed in Microsoft Excel and GraphPad5 software. One-way ANOVA and Tukey's multiple comparison tests were used to determine the statistical significance of binding affinities.

## Antibiotic susceptibility testing—growth curves

Ten milliliters of LB containing 100 mg mL$^{-1}$ ampicillin were inoculated from glycerol stocks of transformed BL21(DE3) *E. coli* containing genes expressing the proteins of interest in pUCP18 vectors. These cultures were incubated at 37 °C with 200 rpm agitation for 16 h. Turbidity was measured through OD600 with a NanoDrop OneC Spectrometer (Thermo Scientific). From the initial cultures, secondary 10 mL cultures were prepared with OD600 values of 0.1. These cultures were grown until OD600 values of 0.4–0.6 were reached and diluted to bring all cultures to the same OD600. Cultures were induced with 1 mM Isopropyl β-D-1-thiogalactopyranoside and incubated for 4 h. Two microliters of culture was added to 198 μL of LB with an appropriate aminoglycoside titration concentration. Each titration point was repeated *n* = 4 times. The formulated 96-well culture plate (Sarstedt) had OD600 measured on a 20-min interval over 17 h using a BioTek

Cytation 5 Cell Imaging Multimode Reader, incubating at 37 °C with 282 cpm orbital agitation. Experiments were repeated with fresh cultures on separate days to ensure consistent trends. Data was analyzed in Microsoft Excel and GraphPad5. The remainder of the culture immediately had its RNA isolated for qPCR experiments.

## Antibiotic susceptibility testing—endpoint MIC assay

Ten milliliters of cultures of LB containing 100 mg mL$^{-1}$ ampicillin were inoculated from glycerol stocks of transformed BL21(DE3) *E. coli* containing genes expressing the proteins of interest in pUCP18 vectors. Experiments were performed using 6xHis-tagged fusion-proteins, as well as untagged wild-type proteins. These cultures were incubated at 37 °C with 200 rpm agitation for 16 h. Turbidity was measured through OD600 with a NanoDrop OneC Spectrometer (Thermo Scientific). From the initial cultures, secondary 10 mL cultures were prepared with OD600 values of 0.1. These cultures were grown until OD600 values of 0.4–0.6 were reached and diluted to bring all cultures to the same OD600. Cultures were induced with 1 mM Isopropyl β-D-1-thiogalactopyranoside and incubated for 4 h. Cultures were diluted to $1 \times 10^6$ cells mL$^{-1}$. Two microliters of culture was added to 198 μL of LB with an appropriate aminoglycoside titration concentration. The formulated 96-well culture plate (Sarstedt) was incubated at 37 °C with 120 rpm orbital agitation. The remainder of the bacterial culture was immediately lysed and processed for Western blotting. OD600 was measured after 20 h using the Molecular Devices Spectra MAX 190 plate reader.

## qPCR

RNA isolation was performed using TRIzol™ (Invitrogen) as per manufacturer guidelines. RNA yield and purity were measured spectrophotometrically using the NanoDrop OneC Spectrometer (Thermo Scientific). RNA was converted to cDNA using the Maxima First Strand cDNA Synthesis Kit (Thermo Scientific) as per manufacturer guidelines. Quantitative real-time reverse transcription PCR (qRT-PCR) was performed using SYBR® Green master mix (Roche) as per manufacturer guidelines and quantified on the QuantStudio 5 instrument. Relative mRNA levels of indicated genes were normalized to the HcaT MFS transporter gene, *hcaT*, which was identified as a suitable reference gene for qPCR experiments involving *Escherichia coli* recombinant protein overexpression[23]. One-way ANOVA with a Tukey multiple comparison test was used to assess statistical significance.

The following primer sequences were used:
*hcaT* forward, GCTGCTCGGCTTTCTCATCC;
*hcaT* reverse, CCAACCACGCTGACCAACC;
AAC(3)-Ia forward, TACGTTCAGGCGGATTACGG;
AAC(3)-Ia reverse, ACTTCTTCACGGATGCCCAG;
AAC(3)-IIIa forward, CGCGTCTGTTAAAGCGGTTG;
AAC(3)-IIIa reverse, CAGAAATTCCGCCAGAACGC;
AAC(3)-XIa forward, CACCACCAACGAAATCCGTG;
AAC(3)-XIa reverse, GTTCATCCAGCATCGCAACC;

## Western blot

Details of Western-blot experiments are provided in Supplementary methods and discussion. Details of antibodies used are provided in Supplementary Table 3. Western blot image provided as Supplementary Fig. 8.

## Data availability

Coordinates and structure factors of all present structures have been deposited in the Protein Data Bank with accession codes 9MGS, 9MGU, 9MGT, 9MH5, 9MH3, 9MH6, and 9MH7. Coordinate files for these structures were also included as Supplementary Data Files 1–7. Sequences of AAC(3)-Ia gene from *A. baumannii* (NCBI: WP_063840259.1), AAC(3)-IIIa gene from *P. aeruginosa* (GenBank: CAA39184.1), and AAC(3)-XIa gene from *C. striatum* (NCBI: NG_050595) were accessed using UniProt[44]. Raw data behind the figures are provided in the Supplementary Data File 8. Additional data

supporting the findings of this study are available from the corresponding author on reasonable request.

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

## Acknowledgements

This research was funded by grants to A.M.B. from the Canadian Institutes for Health Research to A.M.B. (grant PJT-162365) and T.M.S. (PJT-178084). Infrastructure at the McGill University Centre de recherche en biologie structurale, used in this research, is supported by Fonds de Recherche du Québec (Health Sector) Research Centres Grant #288558. Some of the X-ray diffraction data collection described in this paper was performed using the CMCF-BM and CMCF-ID beamlines at the Canadian Light Source, a national research facility of the University of Saskatchewan, which is supported by the Canada Foundation for Innovation, the Natural Sciences and Engineering Research Council, the National Research Council, the Canadian Institutes of Health Research, the Government of Saskatchewan, and the University of Saskatchewan. X-ray diffraction data were also collected at the Advanced Light Source beamline 5.0.1., a DOE Office of Science User Facility under Contract No. DE-AC02-05CH11231, supported in part by the ALS-ENABLE program funded by the National Institutes of Health, National Institute of General Medical Sciences, grant P30 GM124169-01. This research was also enabled in part by support provided by Compute Canada and the Digital Research Alliance of Canada. Additionally, we would also like to thank Lisa Münter and Sidong Huang for generously allowing us access to some of their equipment, and Sara Marshall for consulting on statistics. Funding organizations: CIHR, CFI, FRQS, NRC, NIH, and NSERC.

## Author contributions

Conceived and designed experiments: M.H., M.Z., J.B., K.M., and A.M.B. Small molecule synthesis: A.P. and T.M.S. Performed experiments: M.H., M.Z., J.B., T.G., and D.S.B. Analyzed data: M.H., M.Z., T.G., K.M., and A.M.B. Wrote manuscript: M.H., M.Z., and A.M.B.

## Competing interests

The authors declare no competing interests.
