## [Transparent Peer Review file · Communications Chemistry]

Enzyme-Mediated Aminoglycoside Resistance Without Target Mimicry

Corresponding Author: Professor Albert Berghuis

Version 0:

Reviewer comments:

Reviewer #1

(Remarks to the Author)

In this article, Hemmings M. and collaborators are reporting several novel crystal structures of bacterial N-acetyltransferases (GNAT) of the AAC(3)-Ia and AAC(3)XIa classes. GNAT are essential enzymes for drug resistance as they mediate the acetylation of a broad range of aminoglycosides (AG). GNATs thus impair AG binding to the ribosome and thus trigger AG resistance. The group of Pr. Berghuis leading this study has previously demonstrated and established that AG bind in the GNATs active site in a so-called mimicry mode and in a conformation similar to the one adopted in the ribosomal RNA. Until now all observed AG binding modes in GNATs structures were in that above-described conformation.

In this new study, the authors identified for the first-time a non-canonical mode of binding in two different subclasses of GNAT. The articles described 7 crystal structures in their apo, cofactor and/or ligands bound states. The structures are at medium to high resolution enabling a precise description of the binding mode of the AG.

These structures in comparison with several hundred other X-ray structures of GNAT unambiguously demonstrated an atypical mode of binding of the AG in which the central sugar ring adopts a boat conformation. The structural work is supported by biochemical, in silico, and in vivo data which overall demonstrate that this atypical AG binding is not affecting the efficacy of the AAC enzymes.

Overall, this study is of great interest as it reports an unseen and unclassical binding mode of AG of the extremely well-studied GNAT.

The structural work is very well-conducted and the data are sound. The manuscript is very well written and the discussion is of great interest.

Some modifications are nonetheless required before publication.

Main concerns.

1-Concerning the drug-susceptibility testing, assessing the level of expression of mRNA is not as robust as assessing protein expression. The same mRNA level does not reflect the same protein level as translation efficiency may differ. It is not clear from the materials and methods section if a tag is present or not on the protein, if so, I would recommend doing a western-blot to assess if the proteins have a similar expression level. This is particularly important as there is an absence of correlation between the data from the kinetic and the resistance levels. Maybe the AAC(3)-XIa enzyme is much more expressed than any of the other which could explain the discrepancy between drug-susceptibility testing and enzymatic assays.

2-The drug-susceptibility testing based on growth curves is informative but MICs determination would be even more. As this is a very straightforward experiment to do, I would suggest performing these MICs determinations in liquid media following the standards recommendation.

Minor points:

- Please describe the cloning strategy in the pUCP18 plasmid better.
- AcCoA is more frequently encountered in publication than acCoA as an abbreviation
- In Table I, change Tobramicin to Tobramycin to homogenize with the rest of the text.
- In Table I the angles unit is missing
- line 299, italicize *Pseudomonas aeruginosa*.
- line 303, please give a reference for the LOBSTR *E. coli*
- line 435 italicize *E. coli* as well as in the legend of figure 2 of the supplementary material

-In author contribution T.M.Z. is written instead of T.M.S.

Reviewer #2

(Remarks to the Author)

Modification enzymes are primary cause of transmissible resistance to ribosome-targeting aminoglycoside antibiotics hence understanding their mechanism of action is important to combat AMR threat. In this manuscript, the authors present the structural and functional characterisation of two aminoglycoside-modifying enzymes (AMEs), AAC(3)-Ia and AAC(3)-XIa, which employ an alternative mode of substrate binding & modification relative to their well-characterised homologs. In particular, they find that the enzymes stabilise aminoglycoside molecules in rare boat conformation. It is an intriguing observation as aminoglycosides bind ribosome in low-energy chair conformation, and the authors of the study previously proposed that it is essential for the modifying enzymes to exploit the same conformation to effectively compete with the target ('target mimicry mechanism'). The study thus aims to provide comparative insights into the differences between the newly characterized and target-mimicking enzymes by combining x-ray crystallography and biochemical approaches. I believe this thorough and well-written study will be of interest to Communications Chemistry readership, and in general I support the publication of this work. However, there are some issues that authors need to address before publication is possible:

1. The findings justify better representation & analysis of structural data. Please ensure Figure 1 has better labelling and representations. For example, the figure can include zoom of active sites (insets) clearly showing side chains or surfaces around the bound aminoglycosides and potentially hydrogen bonds with an appropriate labelling. Current Suppl Fig 3 has no labels and cannot compensate for this. Please also clearly label aminoglycosides bound and CoA. Why is the 90 degree arrow in between panels D and G – does it refer to all panels? Panels B, E – are two CoA molecules in different orientations overlaid? It could be also valuable to show electrostatic interactions described in text.
2. The AAC(3)-XIa structures were resolved with only one substrate, suggesting that the current crystallization conditions may not allow for accommodating two substrates. Have the authors explored crystals grown in alternative conditions?
3. To confirm the new conformation of the substrates and to facilitate comparisons between different AMEs discussed below, I strongly recommend at least limited mutational analysis using growth curve, enzyme activity and affinity assays used in the study.
4. AAC(3)-Ia and AAC(3)-XIa exhibit similar K_{cat}/K_M values, yet their behaviours differ significantly in antibiotic susceptibility assays. What might explain this discrepancy?
5. Do substrate conformations correlate with enzyme activity? If so, what is the underlying mechanism? What are the features responsible for affinity and for catalytic efficiency, and can they be dissected by mutational analysis?
6. AAC(3)-XIa has a lower K_{cat}/K_M than AAC(3)-IIIa but shows similar substrate affinity and provides best antibiotic resistance of all enzymes tested. What accounts for these observations? Are AMEs primarily catalytic modifiers or 'molecular sponges'? Are chair enzymes expected to be more catalytically efficient?
7. In a similar vein, AAC(3)-Ia offers no protection despite being able to bind and modify aminoglycoside – why is that, is the protein unstable in *E. coli* or does it have higher activity towards some other aminoglycosides? To eliminate the first possibility, authors should carry out Western blot or SDS-PAGE gel analysis in addition to testing gene expression level.

Very minor comments & typos that might help during revision:

84 crystalize->crystallize

88 The partial-apo protomer of AAC(3)-XIa->do the authors mean the protomer is devoid of ligands? This is confusing as line 76 refers to the whole structure as "partial apo". Do authors mean CoA-bound protomer?

95-97 what is meant by 90 degree shift? unclear in current figure 1

98 pocket->pockets

99 complimentary->complementary

101 10x -> tenfold

107 hinderance->hindrance

108-109: could the different protomers be colored differently to show swapped parts' contribution to ligand binding better?

110: a reader without specific expertise in aminoglycosides like myself might benefit from a scheme of an aminoglycoside molecule with different parts indicated to inform what are the numbering scheme and substituents described in text (similar to the one used in Figure 4).

114-116: these interactions should be shown and labelled clearly in figures

115 what is the ligand stabilisation loop?

122 out of curiosity, does Alphafold predict open structures of AAC(3)-XIa?

125 the discovery maps->experimental maps

130 define 2-DOS ring

137 stabilized->stabilize

150 remove 'novel'

156 target mimicking->target-mimicking

157 are these molecular weights of monomers or dimers?

189-191: this is related to comment 6, as aminoglycosides are catalytically inactivated rather than sequestered they do not strictly compete with the ribosome – they need to work fast enough to inactivate antibiotic?

182-186: could be made more readable by removing some digits

191 transformed *E. coli* - transformed by what?

193 hcaT – italicise

229-230: Might benefit from a figure showing modelling & steric clashes?

260-264 I'm not an expert in this, but entropy discussion is unclear to me. Do I understand correctly that boat- and chair-bound ligands are similarly conformationally restricted by the enzyme? do I understand correctly that higher entropy of boat conformation is compensated by enthalpy of extra hydrogen bonds? Do authors propose now that as boat conformation is still highly accessible in solution, target mimicry idea might not be necessarily useful?

Figures and tables:

Fig.4: In my opinion could be moved to supplementary; I suggest providing comparative structural figures instead
Tables 2 and 3: could be made more readable by removing some insignificant digits

Reviewer #3

(Remarks to the Author)

I co-reviewed this manuscript with one of the reviewers who provided the listed reports. This is part of a Communications Chemistry initiative to facilitate training in peer review and to provide appropriate recognition for Early Career Researchers who co-review manuscripts.

Version 1:

Reviewer comments:

Reviewer #1

(Remarks to the Author)

The authors have addressed all the points that were raised in the first reviewing round.

Dr. M.Blaise

Reviewer #2

(Remarks to the Author)

I am pleased to see the authors have taken substantial efforts to improve their manuscript and figures. I am happy to recommend publication of the revised manuscript.

Reviewer #3

(Remarks to the Author)

It is great to see the manuscript has improved—congratulations on the excellent work.

made.

Reviewer #1

In this article, Hemmings M. and collaborators are reporting several novel crystal structures of bacterial N-acetyltransferases (GNAT) of the AAC(3)-Ia and AAC(3)XIa classes. GNAT are essential enzymes for drug resistance as they mediate the acetylation of a broad range of aminoglycosides (AG). GNATs thus impair AG binding to the ribosome and thus trigger AG resistance. The group of Pr. Berghuis leading this study has previously demonstrated and established that AG bind in the GNATs active site in a so-called mimicry mode and in a conformation similar to the one adopted in the ribosomal RNA. Until now all observed AG binding modes in GNATs structures were in that above-described conformation

In this new study, the authors identified for the first-time a non-canonical mode of binding in two different subclasses of GNAT. The articles described 7 crystal structures in their apo, cofactor and/or ligands bound states. The structures are at medium to high resolution enabling a precise description of the binding mode of the AG

These structures in comparison with several hundred other X-ray structures of GNAT unambiguously demonstrated an atypical mode of binding of the AG in which the central sugar ring adopts a boat conformation. The structural work is supported by biochemical, in silico, and in vivo data which overall demonstrate that this atypical AG binding is not affecting the efficacy of the AAC enzymes

Overall, this study is of great interest as it reports an unseen and unclassical binding mode of AG of the extremely well-studied GNAT.

The structural work is very well-conducted and the data are sound. The manuscript is very well written and the discussion is of great interest.

We would like to thank Reviewer #1 for this very positive assessment of our manuscript. We especially appreciate that Reviewer #1 concludes that “*this study is of great interest*”.

Main suggestions:

- 1 Concerning the drug-susceptibility testing, assessing the level of expression of mRNA is not as robust as assessing protein expression. The same mRNA level does not reflect the same protein level as translation efficiency may differ. It is not clear from the materials and methods section if a tag is present or not on the protein, if so, I would recommend doing a western-blot to assess if the proteins have a similar expression level. This is particularly important as there is an absence of correlation between the data from the kinetic and the resistance levels. Maybe the AAC(3)-XIa enzyme is much more expressed than any of the other which could explain the discrepancy between drug-susceptibility testing and enzymatic assays.*

The constructs used to assess mRNA levels did not contain any tags. This was to avoid introducing additional factors that might impact expression, rationalizing why we did not also include Western-blot studies. However, we agree with Reviewer #1 that mRNA levels may not be an accurate reflection of protein levels. Furthermore, this point was also raised by Reviewer #2 and the Editor strongly encouraged us to include additional experiments. We have thus attempted to use Western-blot studies to assess protein levels. We prepared both

His-tagged and FLAG-tagged versions of our constructs. We subsequently assessed the ability of these tagged proteins to confer resistance. Those experiments showed that the tagged versions behave effectively identical to the untagged proteins. Unfortunately, our Western-blot studies showed that while this protein level was able to provide protection in susceptibility testing, they were not detectable through this method. We dedicated substantial effort to optimize experimental setup, but ultimately had to conclude that this technique is not sufficiently sensitive in this context.

In our revised manuscript we now mention that Western-blot studies were performed but only revealed that protein levels were too low to be detected using this method. Given the negative results of these experiments, we have selected to provide details in supplemental data and not in the main manuscript.

2 The drug-susceptibility testing based on growth curves is informative, but MICs determination would be even more. As this is a very straightforward experiment to do, I would suggest performing these MICs determinations in liquid media following the standards recommendation.

We concur with Reviewer #1 that MIC experiments would add to our manuscript. We have therefore added the MIC experiments. Note that the results from these MIC studies are fully consistent with the data from the growth curve experiments.

Minor suggestions:

1 Please describe the cloning strategy in the pUCP18 plasmid better.

We have updated this section, which we hope has now been clarified.

2 AcCoA is more frequently encountered in publication than acCoA as an abbreviation.

We thank Reviewer #1 for noting this, and we have updated the manuscript accordingly.

3a In Table I, change Tobramicin to Tobramycin to homogenize with the rest of the text.

3b In Table I the angles unit is missing.

We have now corrected these errors in the revised manuscript.

4a line 299, italicize Pseudomonas aeruginosa.

4b line 435 italicize E. coli as well as in the legend of Figure 2 of the supplementary material

We thank Reviewer #1 for noting this, and we have updated the manuscript accordingly.

5 line 303, please give a reference for the LOBSTR E. coli

We have now provided the following reference for the LOBSTR *E.coli* strain:

Andersen, K. R., Leksa, N. C. & Schwartz, T. U. Optimized *E. coli* expression strain LOBSTR eliminates common contaminants from His-tag purification. *Proteins* 81, 1857-1861 (2013). <https://doi.org/10.1002/prot.24364>

5 In author contribution T.M.Z. is written instead of T.M.S.

We must have gotten distracted by some celebrity news. We have now used the correct initials for our collaborator, Prof. Schmeing.

Reviewer #2

Modification enzymes are primary cause of transmissible resistance to ribosome-targeting aminoglycoside antibiotics hence understanding their mechanism of action is important to combat AMR threat. In this manuscript, the authors present the structural and functional characterisation of two aminoglycoside-modifying enzymes (AMEs), AAC(3)-Ia and AAC(3)-XIa, which employ an alternative mode of substrate binding & modification relative to their well-characterised homologs. In particular, they find that the enzymes stabilise aminoglycoside molecules in rare boat conformation. It is an intriguing observation as aminoglycosides bind ribosome in low-energy chair conformation, and the authors of the study previously proposed that it is essential for the modifying enzymes to exploit the same conformation to effectively compete with the target ('target mimicry mechanism'). The study thus aims to provide comparative insights into the differences between the newly characterized and target-mimicking enzymes by combining x-ray crystallography and biochemical approaches. I believe this thorough and well-written study will be of interest to Communications Chemistry readership, and in general I support the publication of this work.

We would also like to thank Reviewer #2 for the supportive evaluation of our manuscript. It is particularly appreciated that Reviewer #2 states that our research findings “*will be of interest to Communications Chemistry readership*” and concludes with expressing support for publication.

Main suggestions:

- 1. The findings justify better representation & analysis of structural data. Please ensure Figure 1 has better labelling and representations. For example, the figure can include zoom of active sites (insets) clearly showing side chains or surfaces around the bound aminoglycosides and potentially hydrogen bonds with an appropriate labelling. Current Suppl Fig 3 has no labels and cannot compensate for this. Please also clearly label aminoglycosides bound and CoA. Why is the 90° arrow in between panels D and G – does it refer to all panels? Panels B, E – are two CoA molecules in different orientations overlaid? It could be also valuable to show electrostatic interactions described in text.*

We have endeavored to fully address this issue. We have updated Figure 1 and supplemental Figure 3, following the suggestion provided. Notably, we hope that Figures 3, 4 and Supplementary Figure 3 now provide a clear view of the aminoglycoside-binding pocket. We have also removed the 90° arrow from Figure 1, hopefully avoiding confusion. Finally, we

have included a figure that show electrostatic properties of the aminoglycoside-binding pockets (Supplementary Figure 4).

2. *The AAC(3)-XIa structures were resolved with only one substrate, suggesting that the current crystallization conditions may not allow for accommodating two substrates. Have the authors explored crystals grown in alternative conditions?*

We have indeed looked for other crystallization conditions using various commercially available crystallization screens. However, following optimization of crystallization parameters, only the reported conditions provided crystals that diffracted to sufficiently high resolution. Additionally, we also attempted soaking of existing crystals with a high molar excess of the aminoglycoside to see if we could force having both aminoglycoside-pockets occupied in the physiological dimer. However, none of these efforts were successful.

3. *To confirm the new conformation of the substrates and to facilitate comparisons between different AMEs discussed below, I strongly recommend at least limited mutational analysis using growth curve, enzyme activity and affinity assays used in the study.*

We are somewhat puzzled by this suggestion. First, it is unclear why additional experiments would be helpful in confirming the different conformation of the aminoglycosides. X-ray crystallographic structure determination is the “gold standard” for determining conformation of molecules, and the reported structures leave no doubt on the boat conformation of the central ring in AAC(3)-Ia and AAC(3)-XIa (see Figure 2). With respect to the suggested mutational analysis, what Reviewer #2 proposes is that we make key mutations in our three enzymes that would alter the enzymes from a boat-binder to a chair-binder, or *vice versa*. For this we would need to alter key residues in the boat-binding AAC(3)-Ia and AAC(3)-XIa enzymes to the *equivalent* residues in the chair-binding AAC(3)-IIIa, and *vice versa*. While in principle, this would be a very worthwhile endeavor, as it may shed light on molecular factors that impact resistance, it is unfortunately not feasible. As Figure 1 and Supplementary Figures 3 and 4 illustrate, the active sites of the three enzymes are dramatically different, such that there are *no* equivalent residues in the aminoglycoside-binding pockets. In fact, only about half of each sequence can be aligned with one another, and for those aligned stretches, the sequence identity is less than 20%. Moreover, these aligned stretches do not coincide with the aminoglycoside-binding pockets. Thus, one cannot simply mutate a chair-binding AAC into a boat-binding AAC, or *vice-versa*, making the proposed mutational studies not feasible.

4. AAC(3)-Ia and AAC(3)-XIa exhibit similar K_{cat}/K_M values, yet their behaviours differ significantly in antibiotic susceptibility assays. What might explain this discrepancy?

We thank Reviewer #2 for noting this. We do not have a real explanation for this, but we speculate that perhaps the following might be at play. Our ITC data indicates that AAC(3)-XIa has a significantly higher affinity for gentamicin and sisomicin than AAC(3)-Ia. This could mean that AAC(3)-XIa is supplementing its catalytic activity by rapidly sequestering

aminoglycosides. However, there are very few cases where sequestration by aminoglycoside resistance enzymes as a mechanism of resistance has been documented, and we want to express caution with this speculation. We have included this speculation in the manuscript with the mentioned caveats.

Note that we have contemplated testing the sequestration hypothesis. However, this is exceedingly difficult if not impossible. Testing this hypothesis would either require an aminoglycoside that retains binding affinity for AAC(3)-XIa but is not a substrate, or an inactive version of AAC(3)-XIa that retains the same affinity for aminoglycosides. For the first approach, a suitable candidate would be a gentamicin or sisomicin analogue that lacks the amino group at the 3 position. However, such variants do not exist and are additionally predicted to lack antibiotic activity. As to creating an inactive AAC(3)-XIa variant, as there are no catalytic residues in AAC(3)-XIa, this would require a variant that is incapable of binding AcCoA while leaving the aminoglycoside binding pocket unaffected. The AcCoA binding site is almost exclusively comprised of backbone interactions, and to our knowledge no inactive GNAT variant has ever been created that lacks AcCoA binding but that retains the overall fold. It is for these fundamental obstacles that we refrained from testing the sequestration hypothesis.

5. Do substrate conformations correlate with enzyme activity? If so, what is the underlying mechanism? What are the features responsible for affinity and for catalytic efficiency, and can they be dissected by mutational analysis?

The question posed here by Reviewer #2 is one aspect of the central question in this manuscript, i.e. whether substrate conformations are correlated with conferring resistance. Prior to the discovery that AAC(3)-Ia and AAC(3)-XIa bind aminoglycosides in never previously observed boat conformation, reported here, this question could not be addressed. Since aminoglycosides were invariably observed to bind in a low-energy conformation, the assumption was that this was essential for conferring resistance, with the mechanism being based on entropic factors, as described in our manuscript. However, the results reported here, demonstrate that life is more complicated as AAC(3)-XIa is able to confer resistance, despite binding the aminoglycoside in the higher-energy boat conformation. As to whether a mutational analysis would be helpful in dissecting factors impacting boat vs chair conformation, as explained above, this is unfortunately not feasible as the enzymes, while belonging to the same GNAT superfamily, are still dramatically different, especially in the aminoglycoside-binding pocket.

6. *AAC(3)-XIa has a lower K_{cat}/K_M than AAC(3)-IIIa but shows similar substrate affinity and provides best antibiotic resistance of all enzymes tested. What accounts for these observations? Are AMEs primarily catalytic modifiers or 'molecular sponges'? Are chair enzymes expected to be more catalytically efficient?*

I hope we have addressed this point in previous responses. The overwhelming consensus is that AMEs, including the ones studied here are principally catalytic modifiers and the reason

for conferring resistance is that the products of the enzyme catalyzed reaction no longer bind to the bacterial ribosome. However, in a few instances, sequestration has been proposed to additionally contribute to the resistance phenotype. We had reasoned that aminoglycoside modifying enzymes, to effectively compete with the ribosome, would also bind the antibiotics in the same low-energy conformation, so as to not incur an entropic penalty. The AAC(3)-Ia and AAC(3)-XIa structures reported here show that there are exceptions to this. The boat-binding enzymes were observed to have significantly lower catalytic efficiency than chair-binding. This was expected, as the boat conformation ligands were stabilised with a greater number of hydrogen bonds. As Reviewer #2 suggests, and we concur, that in this unique situation sequestration might potentially play a role. As mentioned above, we have added some of this discussion in our revised manuscript.

7. In a similar vein, AAC(3)-Ia offers no protection despite being able to bind and modify aminoglycoside – why is that, is the protein unstable in *E. coli* or does it have higher activity towards some other aminoglycosides? To eliminate the first possibility, authors should carry out Western blot or SDS-PAGE gel analysis in addition to testing gene expression level.

This point partially mirrors the first point raised by Reviewer #1. We do however know that AAC(3)-Ia is stable in *E. coli*, as over expression was readily feasible and allowed us to purify ample protein for crystallographic studies. Also note that sisomicin and gentamicin are the only clinically relevant aminoglycosides to which this enzyme confers resistance. With respect to pursuing Western-blot studies please see our response to Reviewer #1. In brief, we indeed pursued this, but protein concentrations were shown to be too low to be detectable by this method.

Minor suggestions:

Reviewer #2 noted the following typos:

- Line 84 *crystalize->crystallize*
- Line 98 *pocket->pockets*
- Line 99 *complimentary->complementary*
- Line 101 *10x -> tenfold*
- Line 107 *hinderance->hindrance*
- Line 125 *the discovery maps->experimental maps*
- Line 137 *stabilized->stabilize*
- Line 150 *remove 'novel'*
- Line 156 *target mimicking->target-mimicking*
- Line 193 *hcaT – italicise*

We have corrected all these typographical errors.

Line 88 The partial-apo protomer of AAC(3)-Xla->do the authors mean the protomer is devoid of ligands? This is confusing as line 76 refers to the whole structure as "partial apo". Do authors mean CoA-bound protomer?

We are sorry for the confusion. In the "partial apo structure: one of the protomers is devoid of ligands, the other protomer of the physiological dimer contains a bound CoA molecule. We have further clarified this in the manuscript now.

Line 95-97 what is meant by 90° shift? unclear in current figure 1.

The view of the two panels is rotated by 90° degrees. However, in hindsight, the ligands landmark the viewpoint such that the 90° arrow is neither necessary nor helpful. We have thus removed this.

Line 108-109 could the different protomers be colored differently to show swapped parts' contribution to ligand binding better?

As requested, we have altered the colours used.

Line 110 a reader without specific expertise in aminoglycosides like myself might benefit from a scheme of an aminoglycoside molecule with different parts indicated to inform what are the numbering scheme and substituents described in text (similar to the one used in Figure 4).

We have added a supplementary figure that provides the information requested by Reviewer #2 (Supplementary Figure 1)

Line 114-116 these interactions should be shown and labelled clearly in figures.

We have made changes to various figures as mentioned in response to Reviewer #2 first comment. These changes also incorporate this point.

Line 115 what is the ligand stabilisation loop?

We realize that in only 3 instances we refer to this region of the AAC(3)-Xla enzyme, which does not justify giving it a specific name and may merely be confusing. Therefore, when discussing this we now specify the hydrogen bond interaction between the aminoglycoside N-3" and O-4" atoms and the Glu111 residue from the adjacent protomer.

Line 122 out of curiosity, does Alphafold predict open structures of AAC(3)-Xla?

We thank the reviewer for this excellent idea. We have checked this, and indeed AlphaFold predicts a more opened conformation (see Supplementary Figure 2).

Line 130 define 2-DOS ring.

Thank you for pointing out this omission, which we have corrected in the revised manuscript.

Line 157 are these molecular weights of monomers or dimers?

This refers to monomers (protomers). This has now been clarified in the manuscript.

Line 189-191 this is related to comment 6, as aminoglycosides are catalytically inactivated rather than sequestered they do not strictly compete with the ribosome – they need to work fast enough to inactivate antibiotic?

Please also see our response to points 4 and 6 raised by Reviewer #2. However, to reiterate, it is the overwhelming consensus that catalytic modification is the primary mode of action of aminoglycoside resistance enzyme. However, in a few cases sequestration is thought to also play a factor. We speculate here that this potentially might also be a factor for AAC(3)-XIa, as it would rationalize some of our biophysical data.

Line 182-186: could be made more readable by removing some digits

As requested, we have removed some of the digits in these tables.

Line 191 transformed *E.coli* - transformed by what?

We understand that referring to *E. coli* strains that were carrying our genes as “transformed”, is not informative. We have thus changed the text to be more explicit.

Line 229-230 Might benefit from a figure showing modelling & steric clashes?

To address this, we have added Supplementary Figure 4.

Line 260-264 I'm not an expert in this, but entropy discussion is unclear to me. Do I understand correctly that boat- and chair-bound ligands are similarly conformationally restricted by the enzyme? Do I understand correctly that higher entropy of boat conformation is compensated by enthalpy of extra hydrogen bonds? Do authors propose now that as boat conformation is still highly accessible in solution, target mimicry idea might not be necessarily useful?

Given the summary on entropy/enthalpy, Reviewer #2 does not give themselves sufficient credit. Indeed, the boat- and chair-bound ligands are both conformationally restricted by the respective enzymes. But to compensate for the entropy penalty of the boat conformation the boat-binding enzymes use additional hydrogen bonds. Given that AAC(3)-XIa is able to confer resistance, despite not employing target mimicry, we cannot escape the conclusion that target mimicry is not a mandatory strategy for aminoglycoside modifying enzymes (this also rationalizes the title of our manuscript). However, given that it has taken more than 25 years to finally find an aminoglycoside modifying enzyme that does not use target mimicry, suggests that this strategy is highly successful and nearly exclusively preferred.

Figure.4 In my opinion could be moved to supplementary; I suggest providing comparative structural figures instead

As suggested, we have added comparative structural figures, specifically Supplementary Figures 3 and 4.

Tables 2 & 3 could be made more readable by removing some insignificant digits

As requested, we have removed some of the digits in these tables (see also above).

Reviewer #3

I co-reviewed this manuscript with one of the reviewers who provided the listed reports. This is part of a Communications Chemistry initiative to facilitate training in peer review and to provide appropriate recognition for Early Career Researchers who co-review manuscripts.

We would like to thank Reviewer #3 for these valuable contributions, and we hope that with all the suggestions provided our manuscript has substantially been improved.

Reviewer #1

In this article, Hemmings M. and collaborators are reporting several novel crystal structures of bacterial N-acetyltransferases (GNAT) of the AAC(3)-Ia and AAC(3)XIa classes. GNAT are essential enzymes for drug resistance as they mediate the acetylation of a broad range of aminoglycosides (AG). GNATs thus impair AG binding to the ribosome and thus trigger AG resistance. The group of Pr. Berghuis leading this study has previously demonstrated and established that AG bind in the GNATs active site in a so-called mimicry mode and in a conformation similar to the one adopted in the ribosomal RNA. Until now all observed AG binding modes in GNATs structures were in that above-described conformation

In this new study, the authors identified for the first-time a non-canonical mode of binding in two different subclasses of GNAT. The articles described 7 crystal structures in their apo, cofactor and/or ligands bound states. The structures are at medium to high resolution enabling a precise description of the binding mode of the AG

These structures in comparison with several hundred other X-ray structures of GNAT unambiguously demonstrated an atypical mode of binding of the AG in which the central sugar ring adopts a boat conformation. The structural work is supported by biochemical, in silico, and in vivo data which overall demonstrate that this atypical AG binding is not affecting the efficacy of the AAC enzymes

Overall, this study is of great interest as it reports an unseen and unclassical binding mode of AG of the extremely well-studied GNAT.

The structural work is very well-conducted and the data are sound. The manuscript is very well written and the discussion is of great interest.

We would like to thank Reviewer #1 for this very positive assessment of our manuscript. We especially appreciate that Reviewer #1 concludes that “*this study is of great interest*”.

Main suggestions:

- 1 Concerning the drug-susceptibility testing, assessing the level of expression of mRNA is not as robust as assessing protein expression. The same mRNA level does not reflect the same protein level as translation efficiency may differ. It is not clear from the materials and methods section if a tag is present or not on the protein, if so, I would recommend doing a western-blot to assess if the proteins have a similar expression level. This is particularly important as there is an absence of correlation between the data from the kinetic and the resistance levels. Maybe the AAC(3)-XIa enzyme is much more expressed than any of the other which could explain the discrepancy between drug-susceptibility testing and enzymatic assays.*

The constructs used to assess mRNA levels did not contain any tags. This was to avoid introducing additional factors that might impact expression, rationalizing why we did not also include Western-blot studies. However, we agree with Reviewer #1 that mRNA levels may not be an accurate reflection of protein levels. Furthermore, this point was also raised by Reviewer #2 and the Editor strongly encouraged us to include additional experiments. We have thus attempted to use Western-blot studies to assess protein levels. We prepared both

His-tagged and FLAG-tagged versions of our constructs. We subsequently assessed the ability of these tagged proteins to confer resistance. Those experiments showed that the tagged versions behave effectively identical to the untagged proteins. Unfortunately, our Western-blot studies showed that while this protein level was able to provide protection in susceptibility testing, they were not detectable through this method. We dedicated substantial effort to optimize experimental setup, but ultimately had to conclude that this technique is not sufficiently sensitive in this context.

In our revised manuscript we now mention that Western-blot studies were performed but only revealed that protein levels were too low to be detected using this method. Given the negative results of these experiments, we have selected to provide details in supplemental data and not in the main manuscript.

2 The drug-susceptibility testing based on growth curves is informative, but MICs determination would be even more. As this is a very straightforward experiment to do, I would suggest performing these MICs determinations in liquid media following the standards recommendation.

We concur with Reviewer #1 that MIC experiments would add to our manuscript. We have therefore added the MIC experiments. Note that the results from these MIC studies are fully consistent with the data from the growth curve experiments.

Minor suggestions:

1 Please describe the cloning strategy in the pUCP18 plasmid better.

We have updated this section, which we hope has now been clarified.

2 AcCoA is more frequently encountered in publication than acCoA as an abbreviation.

We thank Reviewer #1 for noting this, and we have updated the manuscript accordingly.

3a In Table I, change Tobramicin to Tobramycin to homogenize with the rest of the text.

3b In Table I the angles unit is missing.

We have now corrected these errors in the revised manuscript.

4a line 299, italicize Pseudomonas aeruginosa.

4b line 435 italicize E. coli as well as in the legend of Figure 2 of the supplementary material

We thank Reviewer #1 for noting this, and we have updated the manuscript accordingly.

5 line 303, please give a reference for the LOBSTR E. coli

We have now provided the following reference for the LOBSTR *E.coli* strain:

Andersen, K. R., Leksa, N. C. & Schwartz, T. U. Optimized *E. coli* expression strain LOBSTR eliminates common contaminants from His-tag purification. *Proteins* 81, 1857-1861 (2013). <https://doi.org/10.1002/prot.24364>

5 In author contribution T.M.Z. is written instead of T.M.S.

We must have gotten distracted by some celebrity news. We have now used the correct initials for our collaborator, Prof. Schmeing.

Reviewer #2

Modification enzymes are primary cause of transmissible resistance to ribosome-targeting aminoglycoside antibiotics hence understanding their mechanism of action is important to combat AMR threat. In this manuscript, the authors present the structural and functional characterisation of two aminoglycoside-modifying enzymes (AMEs), AAC(3)-Ia and AAC(3)-XIa, which employ an alternative mode of substrate binding & modification relative to their well-characterised homologs. In particular, they find that the enzymes stabilise aminoglycoside molecules in rare boat conformation. It is an intriguing observation as aminoglycosides bind ribosome in low-energy chair conformation, and the authors of the study previously proposed that it is essential for the modifying enzymes to exploit the same conformation to effectively compete with the target ('target mimicry mechanism'). The study thus aims to provide comparative insights into the differences between the newly characterized and target-mimicking enzymes by combining x-ray crystallography and biochemical approaches. I believe this thorough and well-written study will be of interest to Communications Chemistry readership, and in general I support the publication of this work.

We would also like to thank Reviewer #2 for the supportive evaluation of our manuscript. It is particularly appreciated that Reviewer #2 states that our research findings “*will be of interest to Communications Chemistry readership*” and concludes with expressing support for publication.

Main suggestions:

- 1. The findings justify better representation & analysis of structural data. Please ensure Figure 1 has better labelling and representations. For example, the figure can include zoom of active sites (insets) clearly showing side chains or surfaces around the bound aminoglycosides and potentially hydrogen bonds with an appropriate labelling. Current Suppl Fig 3 has no labels and cannot compensate for this. Please also clearly label aminoglycosides bound and CoA. Why is the 90° arrow in between panels D and G – does it refer to all panels? Panels B, E – are two CoA molecules in different orientations overlaid? It could be also valuable to show electrostatic interactions described in text.*

We have endeavored to fully address this issue. We have updated Figure 1 and supplemental Figure 3, following the suggestion provided. Notably, we hope that Figures 3, 4 and Supplementary Figure 3 now provide a clear view of the aminoglycoside-binding pocket. We have also removed the 90° arrow from Figure 1, hopefully avoiding confusion. Finally, we

have included a figure that show electrostatic properties of the aminoglycoside-binding pockets (Supplementary Figure 4).

2. *The AAC(3)-XIa structures were resolved with only one substrate, suggesting that the current crystallization conditions may not allow for accommodating two substrates. Have the authors explored crystals grown in alternative conditions?*

We have indeed looked for other crystallization conditions using various commercially available crystallization screens. However, following optimization of crystallization parameters, only the reported conditions provided crystals that diffracted to sufficiently high resolution. Additionally, we also attempted soaking of existing crystals with a high molar excess of the aminoglycoside to see if we could force having both aminoglycoside-pockets occupied in the physiological dimer. However, none of these efforts were successful.

3. *To confirm the new conformation of the substrates and to facilitate comparisons between different AMEs discussed below, I strongly recommend at least limited mutational analysis using growth curve, enzyme activity and affinity assays used in the study.*

We are somewhat puzzled by this suggestion. First, it is unclear why additional experiments would be helpful in confirming the different conformation of the aminoglycosides. X-ray crystallographic structure determination is the “gold standard” for determining conformation of molecules, and the reported structures leave no doubt on the boat conformation of the central ring in AAC(3)-Ia and AAC(3)-XIa (see Figure 2). With respect to the suggested mutational analysis, what Reviewer #2 proposes is that we make key mutations in our three enzymes that would alter the enzymes from a boat-binder to a chair-binder, or *vice versa*. For this we would need to alter key residues in the boat-binding AAC(3)-Ia and AAC(3)-XIa enzymes to the *equivalent* residues in the chair-binding AAC(3)-IIIa, and *vice versa*. While in principle, this would be a very worthwhile endeavor, as it may shed light on molecular factors that impact resistance, it is unfortunately not feasible. As Figure 1 and Supplementary Figures 3 and 4 illustrate, the active sites of the three enzymes are dramatically different, such that there are *no* equivalent residues in the aminoglycoside-binding pockets. In fact, only about half of each sequence can be aligned with one another, and for those aligned stretches, the sequence identity is less than 20%. Moreover, these aligned stretches do not coincide with the aminoglycoside-binding pockets. Thus, one cannot simply mutate a chair-binding AAC into a boat-binding AAC, or *vice-versa*, making the proposed mutational studies not feasible.

4. AAC(3)-Ia and AAC(3)-XIa exhibit similar K_{cat}/K_M values, yet their behaviours differ significantly in antibiotic susceptibility assays. What might explain this discrepancy?

We thank Reviewer #2 for noting this. We do not have a real explanation for this, but we speculate that perhaps the following might be at play. Our ITC data indicates that AAC(3)-XIa has a significantly higher affinity for gentamicin and sisomicin than AAC(3)-Ia. This could mean that AAC(3)-XIa is supplementing its catalytic activity by rapidly sequestering

aminoglycosides. However, there are very few cases where sequestration by aminoglycoside resistance enzymes as a mechanism of resistance has been documented, and we want to express caution with this speculation. We have included this speculation in the manuscript with the mentioned caveats.

Note that we have contemplated testing the sequestration hypothesis. However, this is exceedingly difficult if not impossible. Testing this hypothesis would either require an aminoglycoside that retains binding affinity for AAC(3)-XIa but is not a substrate, or an inactive version of AAC(3)-XIa that retains the same affinity for aminoglycosides. For the first approach, a suitable candidate would be a gentamicin or sisomicin analogue that lacks the amino group at the 3 position. However, such variants do not exist and are additionally predicted to lack antibiotic activity. As to creating an inactive AAC(3)-XIa variant, as there are no catalytic residues in AAC(3)-XIa, this would require a variant that is incapable of binding AcCoA while leaving the aminoglycoside binding pocket unaffected. The AcCoA binding site is almost exclusively comprised of backbone interactions, and to our knowledge no inactive GNAT variant has ever been created that lacks AcCoA binding but that retains the overall fold. It is for these fundamental obstacles that we refrained from testing the sequestration hypothesis.

5. Do substrate conformations correlate with enzyme activity? If so, what is the underlying mechanism? What are the features responsible for affinity and for catalytic efficiency, and can they be dissected by mutational analysis?

The question posed here by Reviewer #2 is one aspect of the central question in this manuscript, i.e. whether substrate conformations are correlated with conferring resistance. Prior to the discovery that AAC(3)-Ia and AAC(3)-XIa bind aminoglycosides in never previously observed boat conformation, reported here, this question could not be addressed. Since aminoglycosides were invariably observed to bind in a low-energy conformation, the assumption was that this was essential for conferring resistance, with the mechanism being based on entropic factors, as described in our manuscript. However, the results reported here, demonstrate that life is more complicated as AAC(3)-XIa is able to confer resistance, despite binding the aminoglycoside in the higher-energy boat conformation. As to whether a mutational analysis would be helpful in dissecting factors impacting boat vs chair conformation, as explained above, this is unfortunately not feasible as the enzymes, while belonging to the same GNAT superfamily, are still dramatically different, especially in the aminoglycoside-binding pocket.

6. *AAC(3)-XIa has a lower K_{cat}/K_M than AAC(3)-IIIa but shows similar substrate affinity and provides best antibiotic resistance of all enzymes tested. What accounts for these observations? Are AMEs primarily catalytic modifiers or 'molecular sponges'? Are chair enzymes expected to be more catalytically efficient?*

I hope we have addressed this point in previous responses. The overwhelming consensus is that AMEs, including the ones studied here are principally catalytic modifiers and the reason

for conferring resistance is that the products of the enzyme catalyzed reaction no longer bind to the bacterial ribosome. However, in a few instances, sequestration has been proposed to additionally contribute to the resistance phenotype. We had reasoned that aminoglycoside modifying enzymes, to effectively compete with the ribosome, would also bind the antibiotics in the same low-energy conformation, so as to not incur an entropic penalty. The AAC(3)-Ia and AAC(3)-XIa structures reported here show that there are exceptions to this. The boat-binding enzymes were observed to have significantly lower catalytic efficiency than chair-binding. This was expected, as the boat conformation ligands were stabilised with a greater number of hydrogen bonds. As Reviewer #2 suggests, and we concur, that in this unique situation sequestration might potentially play a role. As mentioned above, we have added some of this discussion in our revised manuscript.

7. In a similar vein, AAC(3)-Ia offers no protection despite being able to bind and modify aminoglycoside – why is that, is the protein unstable in *E. coli* or does it have higher activity towards some other aminoglycosides? To eliminate the first possibility, authors should carry out Western blot or SDS-PAGE gel analysis in addition to testing gene expression level.

This point partially mirrors the first point raised by Reviewer #1. We do however know that AAC(3)-Ia is stable in *E. coli*, as over expression was readily feasible and allowed us to purify ample protein for crystallographic studies. Also note that sisomicin and gentamicin are the only clinically relevant aminoglycosides to which this enzyme confers resistance. With respect to pursuing Western-blot studies please see our response to Reviewer #1. In brief, we indeed pursued this, but protein concentrations were shown to be too low to be detectable by this method.

Minor suggestions:

Reviewer #2 noted the following typos:

- Line 84 *crystalize*->*crystallize*
- Line 98 *pocket*->*pockets*
- Line 99 *complimentary*->*complementary*
- Line 101 *10x* -> *tenfold*
- Line 107 *hinderance*->*hindrance*
- Line 125 *the discovery maps*->*experimental maps*
- Line 137 *stabilized*->*stabilize*
- Line 150 *remove 'novel'*
- Line 156 *target mimicking*->*target-mimicking*
- Line 193 *hcaT* – *italicise*

We have corrected all these typographical errors.

Line 88 The partial-apo protomer of AAC(3)-Xla->do the authors mean the protomer is devoid of ligands? This is confusing as line 76 refers to the whole structure as "partial apo". Do authors mean CoA-bound protomer?

We are sorry for the confusion. In the "partial apo structure: one of the protomers is devoid of ligands, the other protomer of the physiological dimer contains a bound CoA molecule. We have further clarified this in the manuscript now.

Line 95-97 what is meant by 90° shift? unclear in current figure 1.

The view of the two panels is rotated by 90° degrees. However, in hindsight, the ligands landmark the viewpoint such that the 90° arrow is neither necessary nor helpful. We have thus removed this.

Line 108-109 could the different protomers be colored differently to show swapped parts' contribution to ligand binding better?

As requested, we have altered the colours used.

Line 110 a reader without specific expertise in aminoglycosides like myself might benefit from a scheme of an aminoglycoside molecule with different parts indicated to inform what are the numbering scheme and substituents described in text (similar to the one used in Figure 4).

We have added a supplementary figure that provides the information requested by Reviewer #2 (Supplementary Figure 1)

Line 114-116 these interactions should be shown and labelled clearly in figures.

We have made changes to various figures as mentioned in response to Reviewer #2 first comment. These changes also incorporate this point.

Line 115 what is the ligand stabilisation loop?

We realize that in only 3 instances we refer to this region of the AAC(3)-Xla enzyme, which does not justify giving it a specific name and may merely be confusing. Therefore, when discussing this we now specify the hydrogen bond interaction between the aminoglycoside N-3" and O-4" atoms and the Glu111 residue from the adjacent protomer.

Line 122 out of curiosity, does Alphafold predict open structures of AAC(3)-Xla?

We thank the reviewer for this excellent idea. We have checked this, and indeed AlphaFold predicts a more opened conformation (see Supplementary Figure 2).

Line 130 define 2-DOS ring.

Thank you for pointing out this omission, which we have corrected in the revised manuscript.

Line 157 are these molecular weights of monomers or dimers?

This refers to monomers (protomers). This has now been clarified in the manuscript.

Line 189-191 this is related to comment 6, as aminoglycosides are catalytically inactivated rather than sequestered they do not strictly compete with the ribosome – they need to work fast enough to inactivate antibiotic?

Please also see our response to points 4 and 6 raised by Reviewer #2. However, to reiterate, it is the overwhelming consensus that catalytic modification is the primary mode of action of aminoglycoside resistance enzyme. However, in a few cases sequestration is thought to also play a factor. We speculate here that this potentially might also be a factor for AAC(3)-XIa, as it would rationalize some of our biophysical data.

Line 182-186: could be made more readable by removing some digits

As requested, we have removed some of the digits in these tables.

Line 191 transformed *E.coli* - transformed by what?

We understand that referring to *E. coli* strains that were carrying our genes as “transformed”, is not informative. We have thus changed the text to be more explicit.

Line 229-230 Might benefit from a figure showing modelling & steric clashes?

To address this, we have added Supplementary Figure 4.

Line 260-264 I'm not an expert in this, but entropy discussion is unclear to me. Do I understand correctly that boat- and chair-bound ligands are similarly conformationally restricted by the enzyme? Do I understand correctly that higher entropy of boat conformation is compensated by enthalpy of extra hydrogen bonds? Do authors propose now that as boat conformation is still highly accessible in solution, target mimicry idea might not be necessarily useful?

Given the summary on entropy/enthalpy, Reviewer #2 does not give themselves sufficient credit. Indeed, the boat- and chair-bound ligands are both conformationally restricted by the respective enzymes. But to compensate for the entropy penalty of the boat conformation the boat-binding enzymes use additional hydrogen bonds. Given that AAC(3)-XIa is able to confer resistance, despite not employing target mimicry, we cannot escape the conclusion that target mimicry is not a mandatory strategy for aminoglycoside modifying enzymes (this also rationalizes the title of our manuscript). However, given that it has taken more than 25 years to finally find an aminoglycoside modifying enzyme that does not use target mimicry, suggests that this strategy is highly successful and nearly exclusively preferred.

Figure.4 In my opinion could be moved to supplementary; I suggest providing comparative structural figures instead

As suggested, we have added comparative structural figures, specifically Supplementary Figures 3 and 4.

Tables 2 & 3 could be made more readable by removing some insignificant digits

As requested, we have removed some of the digits in these tables (see also above).

Reviewer #3

I co-reviewed this manuscript with one of the reviewers who provided the listed reports. This is part of a Communications Chemistry initiative to facilitate training in peer review and to provide appropriate recognition for Early Career Researchers who co-review manuscripts.

We would like to thank Reviewer #3 for these valuable contributions, and we hope that with all the suggestions provided our manuscript has substantially been improved.